# CLARC: C/C++ BENCHMARK FOR ROBUST CODE SEARCH

**Kaicheng Wang**[*]**, Liyan Huang**[*]**, Weike Fang, Weihang Wang**[†]
University of Southern California
`{wangkaic, liyanhua, weikefan, weihangw}@usc.edu`

## ABSTRACT

Efficient code retrieval is critical for developer productivity, yet existing benchmarks largely focus on Python and rarely stress-test robustness beyond superficial lexical cues. To address the gap, we introduce an automated pipeline for code search datasets and present CLARC, a C/C++ benchmark built from real-world GitHub repositories. CLARC contains 1,245 query-code pairs for evaluation and 5,472 pairs for training. The benchmark incorporates LLM-generated natural language queries validated through rigorous human scoring and hypothesis testing. To analyze contextual requirements effectively, our pipeline starts by ensuring code compilability. It then categorizes code snippets by dependency complexity, distinguishing whether the code relies on custom-defined types or helper functions.The pipeline also enables CLARC to stress-test retrieval robustness by introducing challenging settings, including identifier anonymization and compilation to low-level languages like Assembly and WebAssembly. Under these conditions, our evaluation of six state-of-the-art models reveals sharp drops in retrieval effectiveness. The experimental results highlight the models' persistent reliance on lexical features rather than code semantic understanding. Our dataset is publicly available at `https://huggingface.co/datasets/ClarcTeam/CLARC`.

## 1 INTRODUCTION

Code search is becoming indispensable for navigating the increasing size of public codebases (Shekhar, 2024) and fostering software reuse (Di Grazia and Pradel, 2023; Sun et al., 2024). Although recent Large Language Models (LLMs) and Code Language Models (CLMs) demonstrate strong reasoning capabilities for generation tasks (Jiang et al., 2024; Rozière et al., 2024; Hui et al., 2024; Chen et al., 2021), deploying such heavy architectures for large-scale retrieval is often computationally prohibitive (Potvin and Levenberg, 2016; Howell et al., 2023). Searching across extensive repositories demands more efficient methods for compact feature storage and candidate reranking (Liu et al., 2021). Consequently, embedding-based retrieval models remain the standard for scalable navigation, serving as critical components in both standalone search tools and Retrieval-Augmented Generation (RAG) pipelines (Chen et al., 2024; Wang et al., 2025; Zhao et al., 2024).

Various benchmarks and datasets have been developed to evaluate code search systems. However, existing code search benchmarks often suffer from limitations that undermine their practical utility. First, most current datasets (Huang et al., 2021; Husain et al., 2020; Li et al., 2025; Lu et al., 2021; Li et al., 2024; Yao et al., 2018; Heyman and Cutsem, 2020; Yin et al., 2018) prioritize research-favored languages like Python, while neglecting text-to-code tasks of systems programming languages such as C/C++ (Twist et al., 2025) or failing to collect samples from real-world projects (Khan et al., 2024). The imbalance restricts the application of research findings to real-world software development scenarios. Second, many code snippets in current benchmarks lack compilability due to missing include/import statements or helper functions/classes (Cao et al., 2025). The incomplete contexts contrast with the professional practices, where inspecting helper functions and dependencies is crucial for code understanding and analysis. Third, code search benchmarks (Liu et al., 2024a; Khan et al., 2024) rarely evaluate model robustness against textual perturbations, such as variable renaming

---

[*]Equal contribution.
[†]Corresponding author.

(Chen et al., 2025; Qu et al., 2024). The exclusion of these stress tests fails to address the growing reality of code obfuscation and adversarial attacks (Growth Market Reports, 2024; Phylum, 2024; PELock, 2025; Stunnix, 2025; Microsoft, 2025; Faruki et al., 2016; Hort et al., 2025; Yefet et al., 2020), obscuring whether high scores on the code search benchmarks stem from genuine semantic understanding or superficial lexical features.

To address the identified limitations of current code search benchmarks, we introduce CLARC (**C/C++ LA**nguage **R**etrieval with Anonymized **C**ode), a comprehensive dataset comprising 6,717 query-code snippet pairs (1,245 for evaluation and 5,472 for training). CLARC is built through a pipeline designed to mitigate knowledge contamination by extracting real-world snippets from GitHub automatically. The pipeline employs LLMs for query generation, followed by a validation process involving human assessment and rigorous hypothesis testing to ensure query quality. All snippets are fully compilable and categorized by dependency complexity, enabling a nuanced evaluation of how models handle varying levels of contextual information. Moreover, CLARC provides distinct settings that anonymize code identifiers or compile code snippets into low-level languages such as Assembly or WebAssembly (Wasm) (Haas et al., 2017). These settings are specifically designed to isolate the impact of superficial textual features on retrieval accuracy and evaluate models' robustness across different levels of abstraction.

On CLARC, we evaluated six diverse code search methods, including two black-box systems (OpenAI-text-embedding-large, Voyage-code-3), a lightweight encoder model (CodeT5+), a robustness-focused model fine-tuned with augmented data (OASIS), and a model adapted from large-scale CLMs (Nomic-emb-code)). Our experiments reveal a consistent decline in retrieval metrics when identifiers are anonymized, highlighting that the state-of-the-art code search models still rely on superficial lexical features rather than code semantics. We also observe a significant performance drop when snippets are compiled to Assembly or WebAssembly. This degradation underscores the models' limited capability to handle low-level languages.

In summary, our main contributions of this work are:

- Introducing CLARC, a fully compilable C/C++ dataset designed to evaluate retrieval robustness under varying dependency complexity and settings;

- Designing an automated pipeline for scalable benchmark generation, validated through rigorous hypothesis testing, enabling efficient creation of diverse, high-quality evaluation resources that can be reused by other dataset builders; and

- Revealing the models' overreliance on lexical features and their limited capability to generalize to low-level languages.

## 2 RELATED WORKS

**Code Search Models.** Code retrieval has become a critical component of software engineering, facilitating efficient development and improving code quality (Li et al., 2025). Following the paradigm of general dense retrieval (Karpukhin et al., 2020; Izacard et al., 2022; Wang et al., 2024a; Li et al., 2023; Xiao et al., 2024; Bai et al., 2024; Wang et al., 2024b), modern code search models encode the code and queries as embeddings and calculate their similarities. Popular code models, such as CodeBERT (Feng et al., 2020), UniXcoder (Guo et al., 2022), and CodeT5+ (Wang et al., 2023b), have demonstrated significant utility in code search tasks. Subsequently, recent studies have improved the quality of code embedding for retrieval in several directions (Liu et al., 2024b; Gao et al., 2025; Gurioli et al., 2025; Zhang et al., 2024; Nomic Team, 2025; Voyage AI, 2024; OpenAI, 2024). CodeXEmbed (Liu et al., 2024b) proposes a generalizable training approach for code embedding that converts multiple code-related tasks into retrieval tasks. OASIS (Gao et al., 2025) leverages order-based similarity labels to capture semantic nuances. Nomic-emb-code (Nomic Team, 2025) utilizes the CoRNStack dataset (Suresh et al., 2025) and a curriculum-based hard negative mining strategy to boost the model's performance. Closed-source code search models, such as Voyage-code-3 (Voyage AI, 2024) and Open-AI-text-embedding (OpenAI, 2024), also show outstanding results on code retrieval tasks.

**Code Search Benchmarks.** Numerous benchmarks have been developed to evaluate code search models (Husain et al., 2020; Khan et al., 2024; Huang et al., 2021; Lu et al., 2021; Liu et al., 2024a;

Table 1: Statistics of CLARC's evaluation set. LOC stands for lines of code; CC stands for the Cyclomatic Complexity; Src stands for the original code; Asm stands for the Assembly Code, and Wasm stands for the WebAssembly code in `.wat` format. All Code Statistics reported in the table are the average values in the corresponding category. Statistics of the training set are available in Appendix D.1.

| Category | # of Pairs | # of Tokens in Query | Code Statistics | | | | | | |
|---|---|---|---|---|---|---|---|---|---|
| | | | # of Tokens | | | LOC | | | CC |
| | | | Src | Asm | Wasm | Src | Asm | Wasm | Src |
| Group 1 | 526 | 88.3 | 119.2 | 753.7 | 665.5 | 12.8 | 80.7 | 96.2 | 2.9 |
| Group 2 | 469 | 84.7 | 137.7 | 831.3 | 947.1 | 13.3 | 84.4 | 134.4 | 2.8 |
| Group 3 | 250 | 77.4 | 706.9 | 2272.6 | 967.8 | 71.5 | 212.3 | 138.3 | 5.4 |
| Total | 1245 | 84.8 | 244.2 | 1092.7 | 811.4 | 24.8 | 108.9 | 116.1 | 3.4 |

Li et al., 2025; 2024; Yao et al., 2018; Heyman and Cutsem, 2020; Yin et al., 2018). CodeSearchNet challenge (Husain et al., 2020) established an extensive multilingual dataset for semantic code search, while XCodeEval (Khan et al., 2024) built a large executable multilingual benchmark. CoSQA (Huang et al., 2021) and CodeXGLUE (Lu et al., 2021) incorporated real-world user queries, RepoQA (Liu et al., 2024a) focused on understanding long-context code, and COIR (Li et al., 2025) introduced more diverse retrieval tasks and domains. However, these benchmarks have limitations regarding the C/C++ code search. Several neglect C/C++ samples for text-to-code retrieval (Huang et al., 2021; Husain et al., 2020; Li et al., 2025; Lu et al., 2021; Li et al., 2024; Yao et al., 2018; Heyman and Cutsem, 2020; Yin et al., 2018), and others, like XCodeEval (Khan et al., 2024), do not use samples from real-world projects. Furthermore, several benchmarks with C/C++ datasets, such as RepoQA (Liu et al., 2024a), fail to address the influence of superficial textual features. In contrast, CLARC constructs a compilable and extendable C/C++ code search benchmark from real-world GitHub repositories and more deeply evaluates code search models through code anonymization, filling a gap in existing studies.

**LLMs for Benchmarks.** With the rapid advancement of LLMs and their remarkable capabilities, researchers have increasingly utilized these models to help build benchmarks. LLMs help constructing critical evaluation components, including natural language instructions (Zhu et al., 2024), code solutions (Ahmad et al., 2025), and test cases (Schäfer et al., 2024; Alshahwan et al., 2024). They are also applied to support the description generation (Dilgren et al., 2025) and annotation (Sghaier et al., 2025; Liu et al., 2024a; Li et al., 2024; Wang et al., 2023a) of existing datasets. In CLARC, we similarly harness LLMs' ability in code summarization to generate queries for code candidates with hypothesis testing as the validation mechanism, significantly reducing the manual effort required in the benchmark construction process and enhancing the scalability.

## 3 DATASET

This section details the CLARC framework. A high-level overview of the CLARC statistics is presented in Table 1. We first introduce the construction process in Section 3.1, which involves code collection, categorization, the design of specialized robustness settings, and LLM-based query formation. To guarantee the integrity of the dataset, we implement a hypothesis testing process for query validation, which is detailed in Section 3.2.

### 3.1 DATASET CONSTRUCTION

The construction of CLARC follows a four-stage pipeline designed to ensure real-world relevance and rigorous evaluative utility. The process begins with Data Collection, where functions are mined and filtered from popular repositories. These functions are then organized through Categorization based on their reliance on custom types or helper functions. To assess semantic understanding beyond simple lexical matching, we introduce Different Settings, which transform code snippets into

```
bool IsDigit(const char d) {
    return ('0' <= d)&&(d <= '9');
}
```

(a) Group 1

```
int is_set_opt_anc_info (OptAnc* to,
    ↪int anc) {
    if ((to->left & anc) != 0)
        return 1;
    return ((to->right & anc) != 0 ? 1
    ↪ : 0);
}
```

(b) Group 2

```
typedef unsigned char* string;

// Helper function:
int scmp(string s1, string s2 ) {
    ...
}

// Primary function:
void simplesort(string a[], int n, int b) {
    int i, j; string tmp;
    for (i = 1; i < n; i++)
        for (j = i; j > 0 && scmp(a[j-1]+b,
    ↪ a[j]+b) > 0; j--) {
            tmp = a[j];
            a[j] = a[j-1];
            a[j-1] = tmp;
        }
}
```

(c) Group 3

Figure 1: Example Functions of CLARC. The full examples with their corresponding queries can be found in Appendix H.

anonymized or compiled formats. Finally, Query Formation leverages LLMs to generate high-quality natural language descriptions that serve as ground-truth queries for the retrieval task.

### 3.1.1 DATA COLLECTION

We constructed CLARC by mining 144 popular C/C++ repositories from GitHub[1], with a separation between the evaluation set (45 repositories) and the training set (99 repositories). To ensure reproducibility, we first established a compilation environment by creating a whitelist of valid standard library headers used across these projects. We then extracted each function along with its full dependency context, including the call graph and required definitions. At this step, we filtered the dataset to include only functions that can compile within the prepared environment. These filtered functions were then categorized into three groups based on their dependency complexity.

### 3.1.2 CATEGORIZATION

Functions within CLARC were classified into three distinct categories based on their dependencies: Group 1 consists of functions that solely depend on whitelisted standard library functions and types; Group 2 contains functions that rely on standard library functions but utilizes custom-defined variable types; and Group 3 encompasses all other functions that invoke user-defined helper functions. Figure 1 illustrates brief example functions from each group.

To investigate the influence of helper functions on the code search task, we also designed two distinct variants for Group 3: Group 3 Short and Group 3 Long. In Group 3 Short, the primary function and its associated helper functions are treated as separate relevant snippets for retrieval. In contrast, Group 3 Long merges the primary function and its helper functions into a single contiguous code snippet, allowing the models to include both the main logic and its immediate functional dependencies in the context window during retrieval. The details are discussed in Appendix D.3.

### 3.1.3 DIFFERENT SETTINGS

Beyond the standard code search task, CLARC was also designed to evaluate models' ability to comprehend code functionality based on its semantics, rather than relying solely on non-functional lexical features (e.g., class, function, variable names). To facilitate the evaluation under different conditions, we introduced distinct settings of CLARC. The implementation of the settings was detailed in Appendix D.4.

- **Neutralized**: Identifiers in the code snippets are replaced with generic, neutral placeholders like `func_a`, `var_b`, `MACRO_c`, or `class_d`, to reduce non-functional information while preserving the structural role of each identifier.

---

[1]Detailed licensing information is provided in Appendix B.

Table 2: Hypothesis Testing Results. The LLM-generated descriptions for functions in all three groups are comparable or superior in quality to those written by human annotators.

| | LLM Score | LLM 95% CI | Human Score | Human 95% CI | p-value (%) | Avg. Krippendorff's $\alpha$ |
|---|---|---|---|---|---|---|
| Group 1 | 86.0 | (80.5, 91.0) | 60.0 | (52.5, 67.0) | 99.99 | 68.41 |
| Group 2 | 76.5 | (72.5, 80.5) | 72.0 | (67.5, 76.5) | 76.32 | 74.77 |
| Group 3 | 75.5 | (72.0, 79.5) | 71.5 | (67.0, 76.0) | 84.92 | 65.51 |

- **Randomized**: Identifiers in the code snippets are replaced with random names to eliminate all lexical information.
- **Assembly**: Code snippets are compiled to x86 Assembly using `g++`. Most identifiers were eliminated during compilation, while for function names, we removed the symbols via post-processing the Assembly with `objcopy -strip-all`.
- **WebAssembly (Wasm)**: Code snippets are compiled to WebAssembly using Emscripten (Emscripten Team, 2024) with default settings, ensuring no identifiers are preserved.

### 3.1.4 QUERY FORMATION

A time-consuming challenge in constructing natural language to programming language code search benchmarks is obtaining high-quality code descriptions to serve as queries. Our approach utilized LLMs (`o3-mini` and `grok-4`) to automatically generate descriptions for extracted C/C++ functions. The prompts for description generation are provided in Appendix G. The quality of these LLM-generated descriptions was subsequently validated through the hypothesis tests detailed in Section 3.2.

Three manually authored function-description pairs were provided as few-shot examples to guide the desired format and style of the generated queries for all three groups. Furthermore, to enhance the LLM's comprehension of functions in Group 2 and Group 3, we incorporated the functional dependencies, including the definitions of the custom-defined variables and helper functions, into the prompts. As CLARC aims to assess the ability of code search models on code semantics, we explicitly instructed the LLM to avoid including identifier names and generate descriptions based on the code's purpose.

### 3.2 HYPOTHESIS TESTING

To statistically validate the quality of function descriptions generated by LLMs against those from human experts, we adapted the hypothesis testing framework from Wang et al. (2023a). We sampled 125 functions from each category. For each function, we obtained two descriptions: one generated by the LLM and one written by expert software engineers (5+ years of experience). These pairs of descriptions were evaluated by three Computer Science PhD students. To measure inter-annotator agreement, a **shared set** of 50 functions was rated by all three evaluators. The remaining 75 functions were distributed equally, with each student rating a unique disjoint set of 25 functions. This design resulted in a balanced workload of 75 evaluations per student.

**Double-Blind Scoring.** The evaluation followed a strict double-blind protocol. Annotators first verified factual correctness before assessing relative quality. The scoring scheme was defined as follows:

- **Factual Error (-1.0):** Any description containing hallucinations or technical errors received a penalty of -1.0.
- **Correct (+0.5 / +1.0):** If both descriptions were correct, they were compared pairwise:
  - If one description was clearly better, it received **+1.0** and the inferior one received **+0.5**.
  - If both were of equal quality, both received **+0.5**.

**Statistical Analysis and Results.** We compared the aggregate quality of human versus LLM descriptions using bootstrap analysis with $10,000$ iterations. The results are summarized in Table 2.

We measured inter-annotator agreement using Krippendorff's $\alpha$, confirming reliable agreement among the three annotators. Following Wang et al. (2023a), we defined the p-value as the proportion of bootstrap iterations where the total LLM score equaled or surpassed the total human score. Our analysis tested the null hypothesis that the quality of LLM-generated descriptions is comparable to or better than human-generated descriptions. As shown in Table 2, the high p-values across all groups indicate that the LLM consistently matches or outperforms human experts. Furthermore, the 95% confidence intervals for the LLM scores were comparable to, or higher than, those of the human scores. All statistical results validate that the LLM-generated queries achieve a quality level on par with human experts, justifying their use as queries in CLARC.

## 4 EXPERIMENT SETUP

**Models** A small number of embedding models support C/C++, Assembly, and WebAssembly, due to the focus on Python in existing code search research. We evaluated the following models on CLARC in our experiments. The details of the models can be found in Appendix E.

- **BM25** (Trotman et al., 2014): A classical TF-IDF based retrieval algorithm using term frequency, inverse document frequency, and length normalization. It relies on lexical features (e.g., identifier names) and serves as our baseline, and is evaluated only on the standard setting.
- **CodeT5+(110M)** (Wang et al., 2023b): An encoder-decoder Transformer trained on code and text. Its encoder half is used to generate the embeddings for code search.
- **OASIS(1.5B)** (Gao et al., 2025): A code embedding model using the Order-Augmented Strategy with generated hard negatives and order-based similarity labels to learn finer code semantic distinctions.
- **Nomic-emb-code(7B)** (Nomic Team, 2025): A large code embedding model finetuned on CoRNStack (Suresh et al., 2025) using curriculum-based hard negative mining.
- **OpenAI-text-embedding-large** (OpenAI, 2024): A large, closed-source, general-purpose text embedding model. Despite not being code-specific, its broad training enables effective semantic representation of code.
- **Voyage-code-3** (Voyage AI, 2024): A closed-source embedding model optimized for code retrieval, trained on a diverse corpus including extensive code data. It claims state-of-the-art performance on code benchmarks.

**Metrics** We evaluated model performance using standard information retrieval metrics: **NDCG** (Normalized Discounted Cumulative Gain) to assess the quality of ranked lists, **MRR** (Mean Reciprocal Rank) to measure how quickly the first relevant item is found, **MAP** (Mean Average Precision) to gauge overall ranking quality across queries, and **Recall@k** (R@k) to determine the proportion of relevant items retrieved within the top-k results.

## 5 EVALUATION

This section evaluates the code search models across three settings: standard (Section 5.1), neutralized/randomized (Section 5.2), and low-level languages (Section 5.3). A comparison between the standard and neutralized/randomized settings reveals a dramatic performance drop when identifier names are anonymized, indicating that current models rely heavily on lexical information rather than pure code semantics. Furthermore, the poor performance of general-purpose embedding models like OpenAI-text-embedding-large and Voyage-code-3 on low-level languages underscores the need for specialized solutions for retrieval tasks involving Assembly or WebAssembly. Finally, Section 5.4 demonstrates that naive model finetuning is insufficient to resolve these challenges: while finetuning improves general retrieval metrics for some models, the performance gap between standard and neutralized code persists, confirming that standard supervised finetuning cannot overcome the robustness vulnerabilities exposed by this benchmark.

### 5.1 STANDARD SETTING

Table 3 shows model performance on the CLARC standard setting. The limitations of simple text similarity (BM25) and older models like CodeT5+ (early 2023) become clear when compared to

Table 3: Evaluation Results on the Standard Setting. **Bold entries** stand for the maximum values for the metrics in the category. OpenAI stands for OpenAI-text-embedding-large. Voyage stands for Voyage-code-3.

| Model | NDCG | MRR | MAP | R@1 | R@5 | R@10 | R@20 |
|---|---|---|---|---|---|---|---|
| **Group 1** | | | | | | | |
| BM25 | 10.50 | 8.20 | 9.33 | 4.75 | 12.55 | 18.06 | 23.00 |
| CodeT5+ | 64.54 | 58.84 | 59.57 | 47.34 | 74.14 | 82.51 | 89.54 |
| Nomic | 88.61 | 86.23 | 86.41 | 80.04 | **94.11** | 95.82 | 96.96 |
| OASIS | 89.08 | 86.54 | 86.71 | 79.85 | **94.11** | **96.77** | **98.48** |
| OpenAI | 83.57 | 80.16 | 80.45 | 71.67 | 91.06 | 93.92 | 96.01 |
| Voyage | **88.99** | **86.93** | **87.18** | **80.99** | **94.11** | 95.06 | 97.53 |
| **Group 2** | | | | | | | |
| BM25 | 17.83 | 14.64 | 16.42 | 9.81 | 20.47 | 28.36 | 40.72 |
| CodeT5+ | 52.97 | 46.67 | 47.80 | 35.82 | 60.77 | 73.35 | 83.16 |
| Nomic | 93.61 | 91.61 | 91.63 | **86.14** | 98.72 | 99.57 | 99.57 |
| OASIS | 91.11 | 88.30 | 88.33 | 81.02 | 98.29 | 99.57 | **100.00** |
| OpenAI | 85.87 | 81.66 | 81.73 | 71.86 | 95.52 | 98.72 | 99.57 |
| Voyage | **94.06** | **92.10** | **92.11** | 85.93 | **99.57** | **99.79** | **100.00** |
| **Group 3 Short** | | | | | | | |
| BM25 | 10.50 | 11.52 | 7.94 | 2.35 | 7.98 | 11.51 | 15.45 |
| CodeT5+ | 43.55 | 47.82 | 31.24 | 14.68 | 32.44 | 44.83 | 53.93 |
| Nomic | 65.39 | **80.58** | 48.81 | 25.33 | 49.99 | **57.22** | **65.78** |
| OASIS | 63.15 | 73.70 | 47.35 | 25.22 | 48.58 | 56.87 | 62.43 |
| OpenAI | 62.97 | 74.54 | 47.50 | 25.80 | 48.33 | 54.87 | 62.65 |
| Voyage | **66.66** | 80.53 | **50.93** | **27.28** | **51.01** | 57.04 | 64.67 |
| **Group 3 Long** | | | | | | | |
| BM25 | 19.09 | 15.82 | 17.47 | 10.40 | 23.60 | 29.60 | 40.40 |
| CodeT5+ | 21.12 | 17.78 | 19.97 | 12.80 | 26.00 | 32.00 | 50.40 |
| Nomic | 69.46 | 64.93 | 65.66 | 55.20 | 77.20 | 83.60 | 90.00 |
| OASIS | 68.59 | 63.53 | 64.04 | 53.20 | 78.40 | 84.40 | 87.20 |
| OpenAI | 83.80 | 78.76 | 78.83 | 66.40 | 94.00 | 99.20 | **100.00** |
| Voyage | **89.13** | **85.43** | **85.43** | **74.40** | **98.80** | 100.0 | **100.00** |

newer models. Models such as OpenAI-text-embedding-large (early 2024), Voyage-code-3 (late 2024), OASIS, and Nomic-emb-code (2025) demonstrate substantially higher effectiveness. The dominance of the latest models underscores the rapid evolution of code search technology.

Beyond general performance differences, Table 3 also reveals how model performance varies in different CLARC categories. First, the latest models—Nomic-emb-code, OASIS, OpenAI-text-embedding-large, and Voyage-code-3—achieve higher retrieval scores in Group 2 compared to Group 1, suggesting these recent models can effectively utilize custom-defined types for the retrieval task. Additionally, with the exception of CodeT5+, all other models perform better on most retrieval metrics in Group 3 Long than in Group 3 Short. The higher scores imply that the richer contextual information from helper functions in longer code snippets generally enhances code search performance for these models. On the other hand, CodeT5+ displays a contrasting pattern, indicating that it struggles to handle the increased context length and complexity associated with these multiple functions.

To address potential concerns about biases introduced by the LLM-generated queries, we also evaluate the models on the human-annotated subsets. As reported in Appendix F.2, the experimental results on the human-annotated subsets are close to the full dataset results, confirming the quality of the queries in CLARC.

Table 4: Evaluation Results on the Neutralized and Randomized Settings. Neu stands for Neutralized, and Ran stands for Randomized. **Bold entries** stand for the maximum values for the metrics in the category. The evaluation results on the Randomized Setting are the average after 10 trials, and results with standard errors can be found in Appendix F.1.

| Model | NDCG | | MRR | | MAP | | R@1 | | R@5 | |
|---|---|---|---|---|---|---|---|---|---|---|
| | Neu | Ran | Neu | Ran | Neu | Ran | Neu | Ran | Neu | Ran |
| **Group 1** | | | | | | | | | | |
| CodeT5+ | 46.44 | 34.96 | 40.18 | 29.52 | 41.48 | 31.03 | 29.66 | 20.57 | 53.42 | 41.52 |
| Nomic | 87.46 | 77.05 | 84.03 | 72.78 | 84.15 | 73.26 | 76.43 | 63.35 | 93.54 | 85.21 |
| OASIS | 87.13 | 82.33 | 83.66 | 78.74 | 83.78 | 79.02 | **76.62** | 70.11 | 91.44 | 89.62 |
| OpenAI | 74.82 | 66.60 | 70.13 | 60.75 | 70.62 | 61.40 | 59.89 | 48.90 | 84.22 | 76.41 |
| Voyage | **87.56** | **83.85** | **84.22** | **80.68** | **84.33** | **81.00** | 76.05 | **72.66** | **94.87** | **90.53** |
| **Group 2** | | | | | | | | | | |
| CodeT5+ | 19.15 | 14.42 | 15.67 | 11.27 | 17.63 | 12.79 | 10.66 | 6.50 | 22.60 | 16.91 |
| Nomic | 73.37 | 55.27 | 67.65 | 48.23 | 68.14 | 49.24 | 54.80 | 34.75 | 84.65 | 66.74 |
| OASIS | 74.79 | 67.20 | 68.91 | 60.19 | 69.30 | 60.77 | 56.50 | 46.63 | 85.29 | 78.29 |
| OpenAI | 44.20 | 32.45 | 37.14 | 27.55 | 38.53 | 29.11 | 24.95 | 19.21 | 52.88 | 38.51 |
| Voyage | **81.09** | **75.22** | **77.18** | **69.43** | **77.52** | **69.82** | **68.23** | **56.84** | **88.27** | **85.97** |
| **Group 3 Short** | | | | | | | | | | |
| CodeT5+ | 6.52 | 5.73 | 5.37 | 5.59 | 4.56 | 4.28 | 1.33 | 1.40 | 4.82 | 4.13 |
| Nomic | 24.40 | 19.13 | 27.24 | 21.21 | 17.24 | 13.44 | 10.23 | 7.55 | 18.83 | 14.36 |
| OASIS | 27.14 | 25.71 | 29.08 | 29.14 | **19.18** | 17.48 | **11.68** | 10.24 | **21.28** | 19.96 |
| OpenAI | 19.46 | 15.95 | 21.37 | 18.42 | 13.69 | 10.38 | 8.30 | 5.63 | 14.55 | 11.58 |
| Voyage | **27.65** | **30.54** | **31.40** | **35.28** | 18.91 | **20.72** | 11.14 | **12.85** | 20.94 | **23.00** |
| **Group 3 Long** | | | | | | | | | | |
| CodeT5+ | 7.28 | 7.11 | 5.21 | 5.18 | 7.15 | 6.87 | 1.60 | 2.40 | 10.00 | 8.52 |
| Nomic | 38.70 | 30.30 | 34.22 | 26.06 | 35.73 | 27.86 | 26.80 | 19.04 | 44.40 | 34.60 |
| OASIS | 39.35 | 34.69 | 36.08 | 30.51 | 37.65 | 32.15 | 29.20 | 22.96 | 45.20 | 39.00 |
| OpenAI | 34.80 | 33.28 | 29.44 | 28.64 | 30.83 | 30.03 | 20.00 | 20.16 | 42.40 | 39.64 |
| Voyage | **63.90** | **66.40** | **58.58** | **61.15** | **59.45** | **61.95** | **48.40** | **50.48** | **72.80** | **75.04** |

## 5.2 NEUTRALIZED AND RANDOMIZED SETTINGS

Table 4 presents the results of the model evaluation in the neutralized and randomized settings of CLARC. A comparison with the standard setting (Table 3) reveals a substantial and universal decline in performance across all evaluated models. Even top-performing models like Voyage-code-3, Nomic-emb-code, and OASIS experience sharp performance losses, demonstrating that higher baseline effectiveness does not correspond to robustness against identifier anonymization. The pervasive degradation highlights a critical limitation in current code search models: they are heavily dependent on the lexical information from identifiers rather than the underlying code semantics.

In particular, the model performance degrades more severely in the randomized setting. We hypothesize that the relatively higher scores in the neutralized setting stem from residual structural cues preserved in the identifiers (e.g., prefixes distinguishing variables from functions). The randomized setting, however, eliminates these hints and tests the models' comprehension of code semantics without these cues. Among the open-box models, OASIS has the smallest performance reduction, indicating that its training methodology, which incorporates enhanced data for robustness, is beneficial in the neutralized and randomized environment in CLARC.

The retrieval metrics across different groups in the neutralized and randomized settings also deviate from those observed in the standard setting. Specifically, all models now achieve higher performance on Group 1 than on Group 2. The reversal suggests that the models' understanding of custom-defined types (Group 2) relies heavily on the semantic names of the types themselves; when these are obscured, the underlying logic is insufficient for effective retrieval.

Table 5: Evaluation Results on the Assembly and WebAssembly Settings

| Model | NDCG | | MRR | | MAP | | R@1 | | R@5 | |
|---|---|---|---|---|---|---|---|---|---|---|
| | Asm | Wasm | Asm | Wasm | Asm | Wasm | Asm | Wasm | Asm | Wasm |
| Group 1 | | | | | | | | | | |
| OpenAI | 11.50 | 8.89 | 8.61 | 6.60 | 10.12 | 8.29 | 4.25 | 3.22 | 13.51 | 10.30 |
| Voyage | 34.12 | 31.40 | 29.02 | 27.19 | 30.21 | 28.81 | 19.88 | 20.17 | 40.15 | 37.12 |
| Group 2 | | | | | | | | | | |
| OpenAI | 6.86 | 10.85 | 5.15 | 8.14 | 6.70 | 10.11 | 2.40 | 4.20 | 9.15 | 13.09 |
| Voyage | 35.28 | 30.56 | 28.77 | 24.36 | 30.19 | 25.96 | 17.43 | 14.81 | 44.01 | 39.26 |
| Group 3 | | | | | | | | | | |
| OpenAI | 4.79 | 8.90 | 3.46 | 5.86 | 5.04 | 8.14 | 1.60 | 2.63 | 5.20 | 8.77 |
| Voyage | 18.77 | 23.17 | 15.20 | 19.26 | 17.02 | 21.20 | 9.20 | 13.16 | 22.80 | 26.32 |

Additionally, the performance drop is most dramatic in Group 3, indicating a heavy reliance on textual cues for complex code. Notably, the performance gap between Group 3 Short and Group 3 Long widens compared to the standard setting. The larger gap demonstrates that in the absence of descriptive names, the definitions of helper functions (available in Group 3 Long) become essential for inferring the behavior of the primary function.

## 5.3 ASSEMBLY & WEBASSEMBLY SETTINGS

Table 5 presents the performance of models on the Assembly and WebAssembly settings of CLARC. We only evaluate two general-purpose embedding models, OpenAI-text-embedding-large and Voyage-code-3, due to the incompatibility of other models with low-level languages. Additionally, because helper functions must be compiled alongside the primary function in these environments, Group 3 is consolidated into a single configuration rather than separate short and long versions.

Assembly and WebAssembly differ significantly from high-level languages in terms of code length and the vocabulary of unique instructions[2]. When comparing the models' performance in the lower-level language settings to their results in the standard (Table 3) and anonymized settings (Table 4), we observe a more substantial performance drop. The sharp performance decline observed for both OpenAI-text-embedding-large and Voyage-code-3 indicate the models' limited proficiency in understanding these low-level languages. However, the direct comparison between the two models in these challenging low-level language settings reveals that Voyage-code-3 consistently outperforms OpenAI-text-embedding-large. Given that both Assembly and WebAssembly environments inherently remove superficial identifier information, the stronger results from Voyage-code-3 suggest the model possesses better capacity to interpret the program logic.

When comparing the models' performance across different categories within these low-level language settings, Group 1 and Group 2 exhibit broadly comparable results. The similarity suggests that custom-defined types do not introduce substantial retrieval challenges at low-level. In contrast, the models' performance on Group 3 is generally weaker across most metrics. We hypothesize that this difference arises because the dependencies in Group 3, while manageable in high-level languages, translate into much longer and more verbose instruction sequences upon compilation. The increased complexity likely dilutes semantic signals, making the retrieval task considerably more challenging.

## 5.4 MODEL FINETUNING

To investigate whether the robustness challenges identified in our benchmark can be mitigated through supervised adaptation, we finetuned two models, CodeT5+ and OASIS, on our training set. We finetuned each model using InfoNCE loss with in-batch negatives (Gao et al., 2022) for 5 epochs. The performance of the finetuned models on MRR are reported in Table 6.

The results in Table 6 reveal distinct behaviors between the two models after finetuning. CodeT5+ shows MRR improvements across all test settings, suggesting the pre-trained model was insufficiently

---

[2]Comparison details available in Appendix F.3.

Table 6: Performance of the Finetuned Models on MRR. "ft. on X" indicates finetuning on setting X.

| Model | FT Setting | Group 1 | | | Group 2 | | | Group 3 | | |
|---|---|---|---|---|---|---|---|---|---|---|
| | | Std | Neu | Ran | Std | Neu | Ran | Std | Neu | Ran |
| CodeT5+ | W/o ft. | 58.84 | 40.18 | 29.52 | 46.67 | 15.67 | 11.27 | 17.78 | 5.21 | 5.18 |
| | ft. on Std. | 76.51 | 70.19 | 50.71 | 67.00 | 31.93 | 23.80 | 41.91 | 14.97 | 9.14 |
| | ft. on Neu. | 74.49 | 72.69 | 52.34 | 60.62 | 38.66 | 24.08 | 37.90 | 20.44 | 10.25 |
| | ft. on Ran. | 75.89 | 70.02 | 60.93 | 59.98 | 33.82 | 32.03 | 39.52 | 17.23 | 11.70 |
| OASIS | W/o ft. | 86.54 | 83.66 | 78.74 | 88.30 | 68.91 | 60.19 | 63.53 | 36.08 | 30.51 |
| | ft. on Std. | 84.98 | 79.68 | 72.26 | 78.26 | 54.85 | 45.59 | 57.53 | 24.40 | 15.97 |
| | ft. on Neu. | 87.48 | 83.42 | 77.59 | 75.99 | 53.25 | 45.82 | 57.93 | 37.28 | 21.18 |
| | ft. on Ran. | 82.80 | 79.09 | 77.18 | 68.18 | 46.43 | 45.38 | 51.03 | 32.24 | 15.53 |

adapted to C/C++ despite the language being included in its pre-training corpus (Wang et al., 2023b). In contrast, OASIS appears already well-adapted to C/C++, yielding minimal gains from finetuning. Specifically, finetuning OASIS on standard or neutralized data leads to over-specialization—maintaining performance on the target setting while degrading performance on others. Finetuning OASIS on randomized data leads to a worse outcome: an overall degradation in performance.

Crucially, performance gaps between the standard and neutralized/randomized settings persist regardless of the training data used or any performance improvements observed. The persistent performance gap indicates that standard supervised finetuning is insufficient to address the robustness failures in code search. These findings underscore the necessity of benchmarks aiming for robust code search, and we hope our training set can facilitate further research.

# 6  CONCLUSION & FUTURE WORKS

This paper introduces CLARC, a benchmark and automated augmentation pipeline designed to evaluate and enhance code search robustness. Our evaluation reveals that current models perform poorly when textual features are obfuscated or eliminated during compilation, exposing the models' heavy reliance on superficial cues rather than code semantics. Future research can focus on improving model resilience, potentially by leveraging our pipeline to generate large-scale training data for low-level languages like Assembly and WebAssembly, or by establishing new benchmarks targeting robust code search. Finally, we encourage the community to develop more compilable datasets that offer inherent versatility for diverse evaluation and training possibilities.

## ACKNOWLEDGEMENTS

We thank the anonymous reviewers, ACs, and PCs for their constructive review and feedback. This research was supported in part by the U.S. National Science Foundation (NSF) under grants 2409005 and 2321444. Any opinions, findings, and conclusions in this paper are those of the authors only and do not necessarily reflect the views of our sponsors.

## ETHICS STATEMENT

This work introduces CLARC, a benchmark and automated pipeline for evaluating the robustness of natural language code search on C/C++ programs. Our dataset is constructed from publicly available GitHub repositories and we include licensing information and provide license-based splits (Appendix B) to support compliant downstream use. The benchmark does not involve human subject experiments and does not contain personally sensitive information beyond what may already be present in public repositories.

We recognize potential dual-use concerns: improving robustness to identifier anonymization and low-level representations could be used to better analyze obfuscated or malicious code. We mitigate this risk by releasing CLARC for research and evaluation purposes, focusing on retrieval robustness, and by documenting the intended scope and limitations.

We used LLMs to assist query generation and post-writing assistance (Appendix A). We performed human validation of a substantial subset of generated queries and report our validation procedure.

## REPRODUCIBILITY STATEMENT

The code and data required to reproduce the experimental results presented in Section 5 are publicly available. The codebase is hosted on GitHub at `https://github.com/ClarcTeam/CLARC`, and the dataset is available on Hugging Face at `https://huggingface.co/datasets/ClarcTeam/CLARC`. All results were verified to be reproducible with our implementation as of the submission date (September 22, 2025). We note the specific date as certain experimental results rely on API calls (OpenAI-text-embedding-large, Voyage-code-3).

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

## A    THE USE OF LLMS

In this work, the LLMs are used to generate the queries in the dataset, and the quality of the queries is validated by the hypothesis test in Section 3.2. We also use LLMs for post-writing assistance, including proofreading for typographical errors, correcting grammatical errors, and enhancing the clarity of human-authored drafts.

## B    LICENSES

### B.1    DATA SOURCE LICENSE

The GitHub repositories utilized by our dataset have various licensing schemes. While the majority use permissive licenses such as the MIT License, a small subset utilizes relatively restrictive licenses like the GPL. To address potential licensing concerns for users, we tag our dataset samples with corresponding license information and provide separate data splits based on license type, distinguishing between permissive and restrictive licenses.

### B.2    MODEL LICENSES

- **CodeT5+**: BSD 3-Clause License [3]
- **OASIS**: MIT License[4]
- **Nomic-emb-code**: Apache-2.0 [5]
- **OpenAI text-embedding-large**: Users own the embeddings generated by this model according to OpenAI's policies. The linked documentation provides guidance on sharing these embeddings.[6]
- **Voyage-code-3**: Unclear, but we do not include any embeddings from voyage-code-3 in our codebase.

## C    COMPUTE RESOURCE

For the query generation component of CLARC, we utilized OpenAI's `o3-mini` (OpenAI, 2025) and XAI's `grok-4` (xAI, 2025). The combined expense for the prompt engineering, hypothesis testing, and query generation phase was approximately $30.

The evaluation environment for the computational experiments was an x86_64-based system running Ubuntu 22.04. This server was configured with two AMD 48-Core Processors and possessed 1.0 TiB of system RAM. An NVIDIA L40 GPU, featuring 46068 MB of memory, was utilized for the relevant computational tasks. The GPU operated with NVIDIA driver version 550.54.15 and CUDA version 12.4. The aggregate time spent on evaluation across all experiments amounted to roughly 5 GPU hours.

## D    SUPPLEMENTARY DISCUSSION OF THE DATASET

### D.1    TRAINING SET STATISTICS

To facilitate the development of more robust code search models, CLARC also provides a training set of 5,472 query-code pairs. Table 7 presents the statistics for the training set. The training set was constructed using the same automated pipeline described in previous sections from 99 GitHub repositories, disjoint from those used for the benchmark. The training set includes the standard, neutralized, and randomized settings, enabling researchers to fine-tune or train models with improved robustness to identifier variations.

---

[3] https://github.com/salesforce/CodeT5?tab=BSD-3-Clause-1-ov-file
[4] https://huggingface.co/Kwaipilot/OASIS-code-embedding-1.5B
[5] https://huggingface.co/nomic-ai/nomic-embed-code
[6] https://platform.openai.com/docs/guides/embeddings#can-i-share-my-embeddings-online

Table 7: Statistics of the CLARC Training Set. All statistics are average values.

| Category | # of Pairs | # of Tokens in Query | # of Tokens in Code | LOC | CC |
|---|---|---|---|---|---|
| Training Set | 5,472 | 85.6 | 263.8 | 31.2 | 4.6 |

### D.2  DATA FILTERING FOR ASSEMBLY/WEBASSEMBLY

For the x86 Assembly and WebAssembly query-code pairs, we excluded 20 and 227 pairs from the evaluation set, respectively. The exclusions resulted from challenges in function extraction from Assembly/Wasm code. As these exclusions represent only a relatively small fraction of our dataset and do not affect our compilability claims, we consider this acceptable.

### D.3  CATEGORIZATION AND RELEVANCE

We employ two relevance labeling strategies in CLARC to accommodate the varying granularities of the code categories. We first define these strategies and subsequently discuss their impact on specific evaluation metrics.

**Labeling Strategies.**

- **Binary Relevance (Groups 1, 2, and Group 3 Long):** The retrieval task is a standard one-to-one mapping. We assign **Label 1 (Positive)** to the specific function (or merged snippet) the query is based on, and **Label 0 (Negative)** to all other candidates.
- **Graded Relevance (Group 3 Short):** To capture the nuance of dependency retrieval, we differentiate between primary and helper functions. We assign **Label 2 (Highly Relevant)** to the primary function targeted by the query, **Label 1 (Relevant)** to isolated helper functions, and **Label 0 (Irrelevant)** to all others.

**Impact on Evaluation Metrics.**   The choice of labeling strategy influences how performance metrics are calculated:

- **NDCG:** As NDCG measures the ratio between the Discounted Cumulative Gain (DCG) of the model's ranking and the ideal ground truth ranking (IDCG), the order of labels is critical. In the ideal ranking, candidates are ordered such that Label 2 snippets precede Label 1, which precede Label 0.
- **MRR:** In the graded relevance case, Mean Reciprocal Rank focuses on the retrieval of the primary answer. The reciprocal rank is calculated based strictly on the position of **Label 2** snippets; Label 1 snippets are treated as non-target items for this metric.
- **Recall:** Unlike NDCG and MRR, the distinction between graded labels does not influence the Recall calculation. Any code snippet with a non-zero label (Label 1 or 2) is considered relevant, meaning both primary functions and helper functions contribute to the recall score.

### D.4  DATASET SETTINGS

**Neutralized**: Identifiers in the code snippets are replaced with generic, neutral placeholders like `func_a`, `var_b`, `MACRO_c`, or `class_d`, to reduce non-functional information while preserving the structural role of each identifier.

**Randomized**: Identifiers in the code snippets are replaced with random strings. To ensure stability, we performed randomization **ten** times to create corresponding dataset versions, reporting mean results in Table 4. Complete results including standard errors are provided in Appendix F.1.

**Assembly:**   Leveraging the fact that all functions in the benchmark are compilable C/C++ code, we provide the low-level Assembly code generated by compiling the original functions. The objective is to directly assess a model's capability to interpret assembly language instructions and structure. The C/C++ code is compiled to x86 Assembly using the g++ compiler. To achieve a complete

anonymization, we remove the function symbols by post-processing the Assembly using `objcopy –strip-all`.

**WebAssembly (Wasm):** WebAssembly is a binary instruction format that enables secure, fast, and portable execution on the Web (Haas et al., 2017), and has attracted growing attention from the research community (Romano and Wang, 2023a; Perrone and Romano, 2025; Fang et al., 2024; Brito et al., 2022; Baradaran et al., 2024; Lehmann and Pradel, 2022; Huang et al., 2025; Lehmann and Pradel, 2019; Romano et al., 2021; Stiévenart et al., 2022; Wu et al., 2025; Hilbig et al., 2021; Romano and Wang, 2023b; Romano et al., 2022a).

Analogous to the Assembly setting, we first compile functions into WebAssembly binaries by Emscripten (Emscripten Team, 2024), the most widely used WebAssembly compiler (Romano et al., 2022b). These binaries are subsequently converted to the WebAssembly Text Format (`.wat`) using the WABT toolkit (WebAssembly Community, 2025). This setting specifically tests a model's comprehension of WebAssembly code structure and semantics. Compared to the Assembly setting, WebAssembly code features inherent anonymization, as Emscripten does not preserve the function names in the compiled code by default.

# E   MODEL DETAILS

**BM25 (Trotman et al., 2014)** BM25 calculates a relevance score for each function by considering the frequency of query terms within that function (Term Frequency), the inverse frequency of those query terms across the entire code collection (Inverse Document Frequency or IDF), and the function's length relative to the average function length. Since BM25 is based on the superficial features like the identifiers' name, we only use BM25 as the baseline for the standard setting.

**CodeT5+(110M) (Wang et al., 2023b)** CodeT5+ is an encoder-decoder transformer model pre-trained on a vast corpus of source code and associated natural language text. For code search, its encoder generates dense embedding to capture the meaning of both natural queries and functions in programming languages. CodeT5+ is evaluated on the standard, neutralized, and randomized settings.

**OASIS(1.5B) (Gao et al., 2025)** OASIS (Order-Augmented Strategy for Improved code Search) is a code embedding model designed to capture finer semantic distinctions than traditional contrastive learning approaches. It is trained on generated hard-negatives with assigned "order-based similarity labels" to provide a more granular training signal. OASIS learned to generate embeddings that encode a more nuanced understanding of code functionality, aiming to improve code search performance by better discriminating between semantically close but incorrect candidates. OASIS is evaluated in the standard, neutralized, and randomized settings.

**Nomic-emb-code(7B) (Nomic Team, 2025)** Nomic-emb-code is a large-scale embedding model optimized for code retrieval tasks. It utilized the CoRNStack dataset (Suresh et al., 2025) and a curriculum-based hard negative mining strategy, which progressively introduces more challenging negative examples to the model over time using softmax-based sampling during training. Nomic-emb-code has strong code search performance according to its reported state-of-the-art results on benchmarks like CodeSearchNet upon release. Nomic-emb-code is evaluated on the standard, neutralized, and randomized settings.

**OpenAI-text-embedding-large (OpenAI, 2024)** OpenAI's text-embedding-3-large is a large-scale, close-source embedding model accessible via API, widely regarded as a state-of-the-art model for generating general-purpose text representations. While not exclusively trained for code, its training on vast and diverse datasets allows it to produce high-dimensional embeddings that effectively capture semantic meaning for a wide range of inputs, including natural language queries and code snippets. Because of its general-purpose design, we evaluated OpenAI-text-embedding-large on all setting of CLARC.

**Voyage-code-3 (Voyage AI, 2024)** Voyage-code-3 is a specialized, proprietary embedding model explicitly optimized for code retrieval tasks. It is trained on a large, curated corpus combining general text, mathematical content, and extensive code-specific data to handle the nuances of code

Table 8: Evaluation Results on Randomized Setting. **Bold entries** stand for the maximum values for the metrics in the category. Results shown as Mean ± Standard Error after 10 trials.

| Model | NDCG | MRR | MAP | R@1 | R@5 |
|---|---|---|---|---|---|
| **Group 1** | | | | | |
| CodeT5+ | 34.96±1.12 | 29.52±1.21 | 31.03±1.17 | 20.57±1.41 | 41.52±1.88 |
| Nomic | 77.05±0.63 | 72.78±0.78 | 73.26±0.77 | 63.35±1.32 | 85.21±0.51 |
| OASIS | 82.33±0.24 | 78.74±0.33 | 79.02±0.33 | 70.11±0.58 | 89.62±0.49 |
| OpenAI | 66.60±0.70 | 60.75±0.95 | 61.40±0.97 | 48.90±1.47 | 76.41±0.53 |
| Voyage | **83.85±0.44** | **80.68±0.58** | **81.00±0.59** | **72.66±1.06** | **90.53±0.51** |
| **Group 2** | | | | | |
| CodeT5+ | 14.42±0.54 | 11.27±0.49 | 12.79±0.49 | 6.50±0.52 | 16.91±1.13 |
| Nomic | 55.27±1.19 | 48.23±1.32 | 49.24±1.27 | 34.75±1.60 | 66.74±1.80 |
| OASIS | 67.20±0.73 | 60.19±0.82 | 60.77±0.80 | 46.63±1.33 | 78.29±1.42 |
| OpenAI | 32.45±0.65 | 27.55±0.79 | 29.11±0.77 | 19.21±1.14 | 38.51±1.32 |
| Voyage | **75.22±0.54** | **69.43±0.64** | **69.82±0.63** | **56.84±1.18** | **85.97±0.92** |
| **Group 3 Short** | | | | | |
| CodeT5+ | 5.73±0.78 | 5.59±1.04 | 4.28±0.46 | 1.40±0.43 | 4.13±0.69 |
| Nomic | 19.13±0.71 | 21.21±1.01 | 13.44±0.47 | 7.55±0.61 | 14.36±0.53 |
| OASIS | 25.71±0.68 | 29.14±1.29 | 17.48±0.50 | 10.24±0.67 | 19.96±0.62 |
| OpenAI | 15.95±0.63 | 18.42±1.00 | 10.38±0.42 | 5.63±0.56 | 11.58±0.52 |
| Voyage | **30.54±0.36** | **35.28±0.52** | **20.72±0.37** | **12.85±0.63** | **23.00±0.63** |
| **Group 3 Long** | | | | | |
| CodeT5+ | 7.11±0.78 | 5.18±0.69 | 6.87±0.67 | 2.40±0.75 | 8.52±1.53 |
| Nomic | 30.30±1.38 | 26.06±1.14 | 27.86±1.06 | 19.04±1.27 | 34.60±1.97 |
| OASIS | 34.69±0.67 | 30.51±0.78 | 32.15±0.75 | 22.96±1.20 | 39.00±0.85 |
| OpenAI | 33.28±0.74 | 28.64±1.01 | 30.03±1.07 | 20.16±1.56 | 39.64±1.72 |
| Voyage | **66.40±0.50** | **61.15±0.72** | **61.95±0.74** | **50.48±1.56** | **75.04±0.95** |

semantics. Voyage-code-3 demonstrates state-of-the-art performance on a wide suite of code retrieval benchmarks compared to strong generalist models. Similar to OpenAI-text-embedding-large, we also evaluate Voyage-code-3 on all settings of the benchmark.

## F    EVALUATION

### F.1    FULL EVALUATION RESULTS ON RANDOMIZED SETTINGS

As shown in Table 8, the models' performance under the Randomized Setting is stable across 10 trials, with standard errors below 1.0 for most metrics.

### F.2    EVALUATION RESULTS ON HUMAN-ANNOTATED SUBSETS

Our annotation process yielded a subset of 125 queries per group, all authored by expert software engineers with 5+ years of experience. This "LLM-free" subset enables researchers to evaluate retrieval models without potential biases introduced by generative models. Table 9 reports the performance of CodeT5+ and Voyage-code-3 on this human-curated subset compared to the full dataset. The results for Voyage-code-3 are generally consistent across both LLM and human-annotated sets in all three groups. For CodeT5+, while the results on the human set are slightly lower for Group 3, they align well across Groups 1 and 2. Overall, the experimental results on the human set are close to those on the LLM-annotated dataset, confirming the quality of the queries in CLARC.

Table 9: Evaluation Results on Human-Annotated Subsets

| Group | Model | NDCG | MRR | R@1 | R@5 |
|---|---|---|---|---|---|
| Group1 | CodeT5+ | 64.5 | 58.8 | 47.3 | 74.1 |
| Group1 human set | CodeT5+ | 65.4 | 55.6 | 43.4 | 70.4 |
| Group1 | Voyage | 89.0 | 86.9 | 81.0 | 94.1 |
| Group1 human set | Voyage | 91.4 | 89.3 | 83.2 | 96.0 |
| Group2 | CodeT5+ | 53.0 | 46.7 | 35.8 | 60.8 |
| Group2 human set | CodeT5+ | 53.9 | 48.5 | 38.4 | 60.0 |
| Group2 | Voyage | 94.1 | 92.1 | 85.9 | 99.6 |
| Group2 human set | Voyage | 94.0 | 91.9 | 85.6 | 100.0 |
| Group3 short | CodeT5+ | 43.5 | 47.8 | 14.7 | 32.4 |
| Group3 short human set | CodeT5+ | 40.2 | 45.5 | 12.5 | 27.7 |
| Group3 short | Voyage | 66.7 | 80.5 | 27.3 | 51.0 |
| Group3 short human set | Voyage | 65.0 | 81.1 | 25.3 | 48.0 |
| Group3 long | CodeT5+ | 21.1 | 17.8 | 12.8 | 26.0 |
| Group3 long human set | CodeT5+ | 14.2 | 11.6 | 7.2 | 19.2 |
| Group3 long | Voyage | 89.1 | 85.4 | 74.4 | 98.8 |
| Group3 long human set | Voyage | 89.0 | 85.2 | 72.8 | 100.0 |

### F.3 DEEPER ANALYSIS ON LOW-LEVEL LANGUAGE EVALUATION

To investigate why model performance degrades on low-level languages, we analyzed the structural characteristics of code across all three groups in CLARC. Specifically, we measured the average lines of code (LOC), number of tokens, and number of unique keywords or instructions for the Standard, Assembly, and Wasm settings in each category.

As shown in Table 10, compiling to Assembly or WebAssembly results in a 5-10× increase in LOC and a substantial expansion in token usage (e.g., Group 3: $\sim$707 → $\sim$2,273 tokens). Furthermore, the increase in unique instructions indicates a more complex instruction set than standard high-level syntax. The increased complexity in the low-level languages likely overwhelms the models' ability to capture semantic equivalence, as the dense logic of high-level code is diluted across hundreds of low-level instructions.

Table 10: Comparison between Standard, Assembly, and Wasm Settings

| Metric | Group 1 | | | Group 2 | | | Group 3 | | |
|---|---|---|---|---|---|---|---|---|---|
| | Standard | Asm | Wasm | Standard | Asm | Wasm | Standard | Asm | Wasm |
| LOC | 12.8 | 80.7 | 96.2 | 13.3 | 84.4 | 134.4 | 71.5 | 212.3 | 138.3 |
| # of Tokens | 119.2 | 753.7 | 665.5 | 137.7 | 831.3 | 947.1 | 706.9 | 2272.6 | 967.8 |
| # Unique Keywords/Instr. | 4.77 | 16.46 | 13.61 | 4.53 | 15.39 | 13.02 | 6.62 | 21.48 | 15.44 |

### F.4 FINETUNING DETAILS

We follow the contrastive learning paradigm established by Gao et al. (2022). During training, we utilize a batch of $N$ query-code pairs $\{(q_i, c_i)\}_{i=1}^{N}$. For a given query $q_i$, the corresponding code snippet $c_i$ is treated as the positive signal, while the remaining $N-1$ code snippets in the same batch serve as in-batch negatives for the query.

The models are optimized using the InfoNCE loss. For a specific pair $(q_i, c_i)$, the loss $\mathcal{L}_i$ is defined as:

$$\mathcal{L}_i = -\log \frac{\exp(\text{sim}(\mathbf{h}_{q_i}, \mathbf{h}_{c_i})/\tau)}{\sum_{j=1}^{N} \exp(\text{sim}(\mathbf{h}_{q_i}, \mathbf{h}_{c_j})/\tau)} \tag{1}$$

Table 11: Correlation between the embedding shift distance and various features

| Model | Group 1 | | Group 2 | | Group 3 | |
|---|---|---|---|---|---|---|
| | Neu | Ran | Neu | Ran | Neu | Ran |
| *Line of Code* | | | | | | |
| CodeT5+ | −0.113 | −0.131 | −0.115 | −0.196 | −0.025 | 0.080 |
| OASIS | −0.148 | 0.084 | −0.271 | −0.008 | 0.072 | −0.298 |
| Nomic | −0.195 | −0.234 | −0.311 | −0.099 | −0.069 | −0.346 |
| Voyage | −0.308 | −0.321 | −0.341 | −0.390 | −0.303 | −0.275 |
| *Cyclomatic Complexity* | | | | | | |
| CodeT5+ | 0.043 | −0.044 | −0.124 | −0.103 | −0.158 | −0.070 |
| OASIS | −0.081 | −0.113 | −0.117 | 0.141 | 0.017 | 0.081 |
| Nomic | −0.011 | −0.231 | −0.173 | 0.013 | −0.120 | −0.037 |
| Voyage | −0.212 | −0.191 | −0.165 | −0.198 | −0.226 | −0.191 |
| *Perturbation Fraction* | | | | | | |
| CodeT5+ | 0.335 | 0.515 | 0.402 | 0.523 | 0.371 | 0.193 |
| OASIS | 0.762 | 0.342 | 0.596 | 0.632 | 0.375 | 0.369 |
| Nomic | 0.442 | 0.612 | 0.635 | 0.687 | 0.525 | 0.534 |
| Voyage | 0.751 | 0.740 | 0.663 | 0.702 | 0.621 | 0.653 |

where $\mathbf{h}_{q_i}$ and $\mathbf{h}_{c_i}$ are the embedding vectors for the query and code snippet, respectively, $\text{sim}(\cdot, \cdot)$ denotes the cosine similarity, and $\tau$ is a temperature hyperparameter. During our training, we use $\tau = 0.05$ for 5 iterations over the training set for all settings.

### F.5 CORRELATION ANALYSIS ON MODEL ROBUSTNESS

To investigate the determinants of model robustness, we analyze shifts in code embeddings rather than relying on retrieval metrics, which are often confounded by query-candidate similarity. We quantify the embedding shift using the $L_2$ distance between the embedding of the original code ($v_o$) and its perturbed counterparts: the neutralized ($v_n$) and randomized ($v_r$) variants.

We measure the correlation between these shift distances ($\|v_o - v_n\|_2$ and $\|v_o - v_r\|_2$) and three code features: Lines of Code (LOC), Cyclomatic Complexity (CC), and Perturbation Fraction.

As shown in Table 11, structural features (LOC and CC) exhibit weak correlations with embedding shifts. The absolute values of these correlations are predominantly small, suggesting that code length and complexity are not primary drivers of embedding instability. While Voyage-code-3 displays slightly higher sensitivity than other models, the correlations remain weak overall.

In contrast, the Perturbation Fraction emerges as a stronger predictor. Defined as the ratio of the cumulative length of modified identifiers to the total length of the code snippet, the feature exhibits consistent, positive correlations across all groups and models. The larger correlation coefficients support the hypothesis that models' embeddings are driven primarily by superficial lexical similarity rather than deep semantic understanding; as a larger proportion of tokens is modified, even when structural logic is preserved, the resulting embeddings diverge substantially.

# G    QUERY GENERATION PROMPTS

## G.1    PROMPT FOR GROUP 1

Please refer to Figure 2 for the prompt.

Please write a summary for the following C/C++ function that focuses on its functionality without including overly detailed discussions about the specific algorithm or process used. The goal is to ensure that someone who treats the function as a black box can understand its functionality after reading your summary.

Here is the function:
{function_text}

Please read and understand the function step by step. At last, generate your summary after "SUMMARY:". Please note that in your final summary, you should not consider the background of the function, and only focus on the functionality. Also, you should also mention the type of the input and output variables while avoid mentioning the variable names in your final summary.

Figure 2: Prompt for Group 1

## G.2    PROMPT FOR GROUP 2

Please refer to Figure 3 for the prompt.

Please analyze the following function with name {function_name} and generate a concise summary of its functionality. Your summary should:
- Focus solely on what the function does (its functionality) rather than detailing the specific algorithms or processes used.
- Be written from the perspective of a black-box user; that is, someone using the function without needing to know its internal workings.
- Not include any examples or discuss the function's background—only describe its behavior.
- Use high-level language where possible. If a high-level description isn't sufficient, include necessary details.
- Explicitly state the types of the input and output variables (as defined in the provided type declarations) without mentioning any variable names or the function name.

Here are the declaration(s) of the variable types used in the function:
{type_declaration}

Here is the function:
{function_text}

Instructions:
1. Read and understand the function step by step.
2. After your analysis, output your summary on a new line starting with "SUMMARY:"
3. In the final summary, describe only the functionality of the function, explicitly mention the input and output types, while avoiding any reference to variable names, function names, or too much implementation details.

Figure 3: Prompt for Group 2

## G.3 PROMPT FOR GROUP 3

Please refer to Figure 4 for the prompt.

Generate a high-quality description for the following C/C++ function based on the provided guidelines. Focus on summarizing the function's purpose and behavior without reproducing the code or referencing internal variable/function names.

Please follow these guidelines strictly:

1. **Function Summarization**:
   - Do not reproduce the entire function or any code in the description.
   - Focus on summarizing the function's purpose and behavior at a high level.
   - If the snippet includes helper functions or other code, treat them as context to better understand the target function's behavior, but only describe the target function (after the label 'Function to Summarize:') in the summary. This a critical requirement.
   - Explicitly mention the input and output types of the function, while avoiding mentioning specific variable names, function names, or too much implementation details.

2. **Description Quality**:
   - Write clear, concise, and accurate descriptions that avoid unnecessary details.
   - Use high-level descriptions when possible, focusing on what the function does rather than how it does it.
   - If a high-level description is insufficient, include comprehensive details covering all necessary aspects of the function's behavior.
   - Be careful about the details and ensure the description correctly aligns with the function's behavior.

3. **Naming Conventions**:
   - Do not reference internal function or variable names defined within the function body.
   - You may reference names of types, classes, or structs if they are relevant to the description.

4. **Output Format**:
   - Provide the description as plain text.
   - Ensure the description is standalone and does not assume prior context beyond the provided snippet.

5. **Constraints**:
   - Avoid speculative details or assumptions about the code's broader context.
   - Focus only on the functionality implied by the provided snippet.
   - Do not mention any specific identifiers (e.g., variable or function names) unless they are types, classes, or structs.

Your goal is to produce a description that is precise, professional, and aligned with the provided guidelines, suitable for documentation purposes.

Here is the code snippet for description:

{function_text}

Provide the description as plain text, following the guidelines strictly. At last, generate your summary after "SUMMARY:\n". Please note that in your final summary, you should not consider the background of the function, and only focus on the functionality. Also, you should also mention the type of the input and output variables while avoid mentioning the variable names in your final summary.

Here are some examples of how the final summary should look like:

{few_shot_examples}

Figure 4: Prompt for Group 3

### G.4  PROMPT FOR STYLE ALIGNMENT

Please refer to Figure 5 for the prompt. The few-shot examples used in the prompt are sampled from the prompt engineering set.

Your task is to rewrite a summary for a function in C/C++ so that the revised summary is in the same format as the provided examples. Remember, the revised function should not include specific function name or variable names.

Here are the example summaries:

Example Summary 1:
{few_shot_example}

Example Summary 2:
{few_shot_example}

Example Summary 3:
{few_shot_example}

Here is the summary you should rewrite:
SUMMARY: {original_query}

For reference, here is the original function:
{function_text}

Please only output the revised summary. Feel free to include additional details if you think it's helpful to make the format of your revised summary more similar to the examples.

Figure 5: Prompt for Style Alignment

# H  EXAMPLES

## H.1  EXAMPLES FROM GROUP 1

**Query Example**

```
The function takes a single input of type char. It verifies whether the
input character is a numeric digit by checking if it lies between '0'
and '9' (inclusive). If the input character meets this condition, the
function returns true; otherwise, it returns false. The output of the
function is of type bool.
```

**Code Snippet Example**

```c
static bool IsDigit(const char d) {
    return ('0' <= d) && (d <= '9');
}
```

## H.2  EXAMPLES FROM GROUP 2

**Query Example**

```
The function accepts an input of type pointer to a structure (OptAnc)
containing two integers ("left" and "right") and a second input of type
int. It checks whether the int input is represented as a set bit in the
first integer element; if not, it then checks the second integer
element. The output is an int that indicates success (1) if the bit is
set in either of the integer fields, or failure (0) otherwise.
```

**Code Snippet Example**

```c
static int is_set_opt_anc_info(OptAnc* to, int anc) {
    if ((to->left & anc) != 0) return 1;

    return ((to->right & anc) != 0 ? 1 : 0);
}
```

## H.3  EXAMPLES FROM GROUP 3

**Query Example**

```
This function performs an in-place sort on an array of unsigned char
pointers, which represent strings, for a specified number of elements
starting from the beginning of the array. It orders the strings in
ascending lexicographical order based on the substrings beginning at a
given offset position in each string. The function takes an array of
unsigned char pointers, an integer specifying the number of elements to
sort, and an integer offset for comparisons, and returns void.
```

**Code Snippet Example**

```c
typedef unsigned char* string;

int scmp( unsigned char *s1, unsigned char *s2 )
{
    while( *s1 != '\0' && *s1 == *s2 )
    {
        s1++;
        s2++;
    }
```

```
    return( *s1-*s2 );
}

static void simplesort(string a[], int n, int b)
{
   int i, j;
   string tmp;

   for (i = 1; i < n; i++)
      for (j = i; j > 0 && scmp(a[j-1]+b, a[j]+b) > 0; j--)
         { tmp = a[j]; a[j] = a[j-1]; a[j-1] = tmp; }
}
```

