# OpenReview forum: "CLARC: C/C++ Benchmark for Robust Code Search"
_ICLR.cc/2026/Conference — ICLR 2026 Poster_

### Official Review · Reviewer_eTqq · 2025-10-28

**Soundness:** 3
**Presentation:** 2
**Contribution:** 2
**Rating:** 4
**Confidence:** 5

**Summary:**

CLARC introduces a comprehensive C/C++ code search benchmark addressing the limitations in existing benchmarks. The benchmark contains 1,245 fully compilable query-code pairs sourced from popular GitHub repositories, categorized into three groups based on dependency complexity. Unlike previous benchmarks that focus primarily on Python, CLARC systematically evaluates model C/C++ code search through multiple settings: standard code, neutralized identifiers (generic placeholders), randomized identifiers, and low-level representations (Assembly/WebAssembly). The authors developed an automated pipeline using LLMs to generate natural language queries, validated through hypothesis testing against human expert annotations. The results demonstrate that current models perform poorly on low-level languages and lack robust understanding of code functionality.

**Strengths:**

1. **Fills gap for C/C++ retrieval**: First comprehensive C/C++ robustness benchmark addressing the field's Python bias with real-world compilable code
2. **Systematic robustness evaluation**: Structured testing across identifier anonymization and compilation settings isolates semantic understanding from lexical pattern matching
3. **Scalable automated methodology**: LLM-based query generation with statistical validation enables cost-effective benchmark expansion while reducing knowledge contamination

**Weaknesses:**

1. **Questionable prevalence of target language**
The paper claims C/C++ represents "industrially prevalent languages," but this assertion lacks supporting evidence. While C/C++ has importance in systems programming, languages like Java, JavaScript, Python, and C# arguably have broader industrial adoption across web development, enterprise applications, and data science. The authors should justify why C/C++ specifically addresses an industrial need, or broaden their claim to acknowledge the more diverse landscape of industrially relevant languages.

2. **Questionable use cases of benchmark setting**
The practical necessity of code retrieval across different abstraction levels is questionable. The anonymized identifier setting (func_a, var_b, etc.) represents an artificial scenario, which is contrasted by most professional developers that will follow naming conventions and write meaningful identifiers. Similarly, searching Assembly or WebAssembly code has limited real-world utility, as most developers work exclusively in high-level languages and rely on compilers for low-level translation.

3. **Limited Dataset Scale and Missing Training Components**
Despite introducing an automated pipeline, the benchmark contains only 1,245 query-code pairs, which is relatively small for modern machine learning evaluation. More critically, while the authors demonstrate automated dataset construction capabilities, they provide no training set. Given the automated nature of their pipeline, supplying training data would significantly enhance the benchmark's practical value for model development.

4. **Insufficient Validation of LLM-Generated Queries**
The quality assessment of LLM-generated descriptions relies on only 125 samples per category, which may not be representative of the full dataset's quality. More fundamentally, LLM-generated queries tend toward stylistic homogeneity, lacking the diversity found in real-world project descriptions and developer queries. This limitation reduces the benchmark's ecological validity and may not adequately test model robustness against natural query variation.

5. **Poor Results Presentation**
The experimental results are poorly organized across Tables 3-5, requiring excessive cross-referencing to compare model performance across different settings. A consolidated presentation showing all experimental conditions for each model/group combination would significantly improve readability and facilitate comparative analysis.

**Questions:**

While the work addresses the gap (lacking C/C++) in code search evaluation, the authors should reconsider the practical relevance of their chosen scenarios and significantly expand both the dataset size and validation methodology.

---

> ### Author Response · Authors · 2025-11-20
>
> We sincerely thank the reviewer for the thoughtful and constructive feedback. We respond to each point below.
>
> ## Dataset extension and model finetuning
>
> We acknowledge the reviewer's feedback regarding the practical value of providing a training set. In response, we utilized our automated pipeline to generate a training dataset from 99 additional C/C++ repositories, yielding approximately 5,400 (query, code snippet) pairs. We report the statistics of the training set in Table 1, and the training set will be added to our dataset.
>
> **Table 1:**
> | # of Samples | # of Tokens in Query | # of Tokens in Code | LOC  | CC  |
> |--------------|----------------------|---------------------|------|-----|
> | 5472         | 85.6                 | 263.8               | 31.2 | 4.6 |
>
>
> To validate the utility of this newly generated data, we also finetuned CodeT5+ and OASIS using InfoNCE loss with in-batch negatives[1] for 5 epochs, and the performance of the finetuned models are reported in Table 2.
>
> **Table 2: Performance of the Finetuned Models. All reported metrics are Mean Reciprocal Rank (MRR) in percentage due to space constraints.**
>
> | | | | Group 1 |  | |  | Group 2 |  | |  | Group 3 |  |
> |---|---|:---:|:---:|:---:|---|:---:|:---:|:---:|---|:---:|:---:|:---:|
> | | FT Setting | Standard | Neutralized | Randomized | | Standard | Neutralized | Randomized | | Standard | Neutralized | Randomized |
> | **CodeT5+** | W/o ft. | 58.8 | 40.2| 29.5| | 46.7| 15.7| 11.3| | 17.8| 5.21| 5.18|
> | | ft. on Std. | 76.5 | 70.2 | 50.7 | | 67.0 | 31.9 | 23.8 | | 41.9 | 15.0 | 9.1 |
> | | ft. on Neu | 74.5 | 72.7 | 52.3 | | 60.6 | 38.7 | 24.1 | | 37.9 | 20.4 | 10.3 |
> | | ft. on Ran. | 75.9 | 70.0 | 60.9 | | 60.0 | 33.8 | 32.0 | | 39.5 | 17.2 | 11.7 |
> | **OASIS** | W/o ft. | 86.5| 83.7| 78.7| | 88.3| 68.9| 60.2| | 63.5| 36.1| 30.5|
> | | ft. on Std. | 85.0 | 79.7 | 72.3 | | 78.3 | 54.9 | 45.6 | | 57.5 | 24.4 | 16.0 |
> | | ft. on Neu | 87.5 | 83.4 | 77.6 | | 76.0 | 53.3 | 45.8 | | 57.9 | 37.3 | 21.2 |
> | | ft. on Ran. | 82.8 | 79.1 | 77.2 | | 68.2 | 46.4 | 45.4 | | 51.0 | 32.2 | 15.5 |
>
>
> The results reveal distinct behaviors between CodeT5+ and OASIS following finetuning. CodeT5+ demonstrates significant MRR improvements across all test settings (standard, neutralized, and randomized). These results indicate that the pretrained CodeT5+ remained insufficiently adapted to C/C++, despite exposure to the language during pre-training [2], and our fine-tuning process effectively addresses this deficiency. Conversely, OASIS appears already well-adapted to C/C++ code; thus, finetuning yields minimal gains. In fact, when finetuning on standard or neutralized data, OASIS maintains performance comparable to the pretrained checkpoint on the target setting but degrades on others. Notably, finetuning on the randomized setting leads to overall performance degradation for OASIS.
>
> Crucially, the distinct performance gaps between the standard and neutralized/randomized setting remain. Regardless of the training data used or the performance boosts observed, models consistently underperform on neutralized and randomized settings compared to the standard setting. This persistence indicates that improving code search robustness is a non-trivial task that cannot be solved simply by exposing models to more training data. These findings underscore the necessity of our proposed benchmark to rigorously measure these vulnerabilities. We will release this training set alongside the benchmark to facilitate further research into these robustness challenges.
>
> [1] Tianyu Gao, Xingcheng Yao, and Danqi Chen. 2021. SimCSE: Simple Contrastive Learning of Sentence Embeddings. In Proceedings of the 2021 Conference on Empirical Methods in Natural Language Processing
>
> [2] Wang, Yue, et al. "Codet5+: Open code large language models for code understanding and generation." Proceedings of the 2023 conference on empirical methods in natural language processing. 2023.

---

> > ### Comment · Reviewer_eTqq · 2025-11-27
> > **Reply to authors**
> >
> > Dear authors,
> >
> > 1. thanks for providing training set and finetuning. this make the dataset more applicable. (concern resolved)
> > 2. thanks for consolidating the experimental results. (concern resolved)
> > 3. regarding the benchmark setting, the rebuttal is weak and cannot convince me, please provide more concrete use cases where user need to retrieval c/c++ code in anonymized and randomized forms. (concern pending to be resolved)
> > 4. regarding LLM-Generated queries, as comprehensive benchmark, you need to make sure all your 1,245 queries are of good quality and manually examined, rather than relying on LLMs. (concern pending to be resolved)
> >
> > Hence, i will keep my score.

---

> > > ### Author Response · Authors · 2025-12-01
> > >
> > > We thank the reviewer for the reply. We respond to your remaining concerns below:
> > >
> > > ## Remaining Concern 1: ​​Please provide more concrete use cases where user need to retrieval c/c++ code in anonymized and randomized forms.
> > >
> > > We highlight the critical value of our anonymization settings in CLARC based on three factors:
> > >
> > > * **Industrial Relevance**: Code obfuscation tools [1-4] are widely adopted in both industry and research for code protection [5]. Consequently, users of obfuscation tools work with code where identifiers are randomized strings, yet still need to effectively retrieve and utilize such code. In these scenarios, code search models must understand semantics beyond superficial identifier names.
> > >
> > > * **Real-World Coverage**: Obfuscated code is prevalent "in the wild." Research indicates that a considerable number of packages and snippets are obfuscated [5, 6]. A model capable of handling CLARC's randomized setting can effectively utilize this broader corpus, providing more comprehensive results to users.
> > >
> > > * **Adversarial Robustness**: Identifiers are frequently exploited in adversarial attacks against code embedding models [7, 8], where attackers randomize names to deceive the system. Our anonymized and randomized settings serve as robustness evaluations, testing whether models can maintain semantic understanding when informative variable names are unavailable, which is a critical step toward mitigating vulnerability to adversarial attacks.
> > >
> > > [1] PELock, https://www.pelock.com/products/pelock
> > >
> > > [2] Stunnix, http://stunnix.com/
> > >
> > > [3] Android Code Protection via Obfuscation Techniques: Past, Present and Future Directions
> > >
> > > [4] "Use Dotfuscator Community to Protect .NET Apps." Microsoft Learn, Microsoft, 12 Mar. 2025, learn.microsoft.com/en-us/visualstudio/ide/dotfuscator/?view=visualstudio.
> > >
> > > [5] Code Obfuscation Tools Market Research Report 2033, https://growthmarketreports.com/report/code-obfuscation-tools-market.
> > >
> > > [6] Q3 2024 Evolution of Software Supply Chain Security Report, https://blog.phylum.io/q3-2024-evolution-of-software-supply-chain-security-report/.
> > >
> > > [7] Yefet, Noam, Uri Alon, and Eran Yahav. "Adversarial examples for models of code." Proceedings of the ACM on Programming Languages 4. OOPSLA (2020): 1-30.
> > >
> > > [8] Hort, Max, Linas Vidziunas, and Leon Moonen. "Semantic-Preserving Transformations as Mutation Operators: A Study on Their Effectiveness in Defect Detection." 2025 IEEE International Conference on Software Testing, Verification and Validation Workshops (ICSTW). IEEE, 2025.

---

> > > > ### Author Response · Authors · 2025-12-01
> > > >
> > > > ## Remaining Concern 2: LLM-generated query quality
> > > >
> > > > First, we emphasize that the query quality in CLARC is rigorously validated in Section 3.4. Our human validation covers over 30% of the data points, a higher proportion than most prior works involving LLM-constructed datasets [9, 10, 11].
> > > >
> > > > Additionally, our annotation process for hypothesis testing allowed us to create a high-quality subset of CLARC where all queries were authored exclusively by expert-level software engineers with 5+ years of experience. This "LLM-free" subset allows researchers to minimize potential biases from generative models. Below, we report the performance of CodeT5+ and Voyage-code-3 on this human-curated set compared to the full dataset. (Full evaluations for all settings will be included in the final draft's appendix.)
> > > >
> > > > **Table 3**
> > > > | Group | Model | NDCG | MRR | R@1 | R@5 |
> > > > |-------|-------|------|-----|-----|-----|
> > > > | Group1 | CodeT5+ | 64.5 | 58.8 | 47.3 | 74.1 |
> > > > | Group1 human set | CodeT5+ | 65.4 | 55.6 | 43.4 | 70.4 |
> > > > | Group1 | Voyage | 89.0 | 86.9 | 81.0 | 94.1 |
> > > > | Group1 human set | Voyage | 91.4 | 89.3 | 83.2 | 96.0 |
> > > > |  |  |  |  |  |  |
> > > > | Group2 | CodeT5+ | 53.0 | 46.7 | 35.8 | 60.8 |
> > > > | Group2 human set | CodeT5+ | 53.9 | 48.5 | 38.4 | 60.0 |
> > > > | Group2 | Voyage | 94.1 | 92.1 | 85.9 | 99.6 |
> > > > | Group2 human set | Voyage | 94.0 | 91.9 | 85.6 | 100.0 |
> > > > |  |  |  |  |  |  |
> > > > | Group3 short | CodeT5+ | 43.5 | 47.8 | 14.7 | 32.4 |
> > > > | Group3 short human set | CodeT5+ | 40.2 | 45.5 | 12.5 | 27.7 |
> > > > | Group3 short | Voyage | 66.7 | 80.5 | 27.3 | 51.0 |
> > > > | Group3 short human set | Voyage | 65.0 | 81.1 | 25.3 | 48.0 |
> > > > |  |  |  |  |  |  |
> > > > | Group3 long | CodeT5+ | 21.1 | 17.8 | 12.8 | 26.0 |
> > > > | Group3 long human set | CodeT5+  | 14.2 | 11.6 | 7.2 | 19.2 |
> > > > | Group3 long | Voyage | 89.1 | 85.4 | 74.4 | 98.8 |
> > > > | Group3 long human set | Voyage  | 89.0 | 85.2 | 72.8 | 100.0 |
> > > >
> > > > As shown in Table 3, the models exhibit comparable performance across both the full CLARC dataset and the human-only subset. This consistency demonstrates that our LLM-generated queries are of comparable quality to those written by human experts.
> > > >
> > > >
> > > > [9] Li, Rui, et al. "Optimizing code retrieval: High-quality and scalable dataset annotation through large language models." Proceedings of the 2024 Conference on Empirical Methods in Natural Language Processing. 2024.
> > > >
> > > > [10] Wang, Jianyou Andre, et al. "Scientific document retrieval using multi-level aspect-based queries." Advances in Neural Information Processing Systems 36 (2023): 38404-38419.
> > > >
> > > > [11] Wu, Siwei, et al. "Scimmir: Benchmarking scientific multi-modal information retrieval." Findings of the Association for Computational Linguistics: ACL 2024.

---

> ### Author Response · Authors · 2025-11-20
>
> ## Industrial relevance of C/C++
>
> We thank the reviewer for raising this question. While languages such as JavaScript and Python dominate web and enterprise software, C/C++ remain the industry standard for critical infrastructure, including operating systems, embedded devices, and high-performance computing [3,4]. Despite this importance, C/C++ benchmarks for code search remain scarce compared to higher-level languages, which we highlighted in the introduction of our paper. Additionally, C/C++ uniquely supports compilation to Assembly and Wasm, enabling robustness studies across multiple abstraction levels - capabilities not directly available for languages like Python or JavaScript.
>
> Furthermore, the benchmark construction pipeline introduced in our work can serve as a template for creating benchmarks for other PLs with broad industry adoption. This process involves crawling source code, performing manual annotation and hypothesis testing, and generating the final dataset.
>
> We will revise our paper to more clearly reflect this nuanced industrial landscape.
>
> [3] Stack Overflow. "Technology." 2025 Stack Overflow Developer Survey, Stack Overflow, 29 July 2025, https://survey.stackoverflow.co/2025/technology.
>
> [4] Jansen, Paul. "TIOBE Index for November 2025." TIOBE, TIOBE Software, Nov. 2025, https://www.tiobe.com/tiobe-index/.
>
>
> ## Clarification on the practicality of benchmark settings
> Firstly, we would like to clarify that code anonymization and obfuscation with identifier renaming are widely adopted practices in both the research community and commercial software protection tools [5, 6, 7, 8]. Our anonymized settings therefore represent a realistic scenario for evaluating current models' ability to understand obfuscated code, which underscores the practical relevance of our benchmark.
>
> Besides, we emphasize that identifiers are frequently exploited in adversarial attacks against code embedding models [9, 10]. Our anonymized and randomized settings can also serve as a critical robustness stress test, designed to evaluate the model's semantic understanding when informative variable names are unavailable. Furthermore, these settings provide an intermediate stage between high-level C/C++ code and low-level assembly or WASM.
>
> We will add more references on adversarial attacks against Code Language Models (CLMs) to the related works section in our revised draft.
>
> [5] PELock, https://www.pelock.com/products/pelock
>
> [6] Stunnix, http://stunnix.com/
>
> [7] Android Code Protection via Obfuscation Techniques: Past, Present and Future Directions
>
> [8] "Use Dotfuscator Community to Protect .NET Apps." Microsoft Learn, Microsoft, 12 Mar. 2025, learn.microsoft.com/en-us/visualstudio/ide/dotfuscator/?view=visualstudio.
>
> [9] Yefet, Noam, Uri Alon, and Eran Yahav. "Adversarial examples for models of code." Proceedings of the ACM on Programming Languages 4. OOPSLA (2020): 1-30.
>
> [10] Hort, Max, Linas Vidziunas, and Leon Moonen. "Semantic-Preserving Transformations as Mutation Operators: A Study on Their Effectiveness in Defect Detection." 2025 IEEE International Conference on Software Testing, Verification and Validation Workshops (ICSTW). IEEE, 2025.

---

> > ### Author Response · Authors · 2025-11-20
> >
> > ## Validation of LLM-Generated queries
> >
> > To our knowledge, there is a scarcity of publicly available, detailed pairs of natural language queries and corresponding C/C++ code snippets. Other works also mention this issue [11, 12]. Our experiments demonstrate that the quality of LLM-generated descriptions is comparable to that of human experts. We believe the queries in our benchmark are of higher quality than typical crowdsourced outputs.
> >
> > Regarding the reviewer’s concern about the validation set size, we would like to clarify that our human validation set has covered over ~30% of all data points in the current version of CLARC. To ensure robustness, we strategically focused our annotation efforts on the most challenging data, with 50% of all points in Group 3 being manually validated. This level of validation is consistent with, and in some cases exceeds, the standards in recent, related work. For instance, [13] manually assessed 200 samples for a similar code retrieval task focusing on Python, while broader LLM-annotated tasks [14, 15] used 250 and 100 samples, respectively, on datasets larger than CLARC.
> >
> > To address the concern about stylistic homogeneity, we will tag the human-written subset distinctly for users who prefer natural queries.
> >
> > [11] Liuwen Cao, Yi Cai, Jiexin Wang, Hongkui He, and Hailin Huang. 2024. Beyond Code: Evaluate Thought Steps for Complex Code Generation. In Proceedings of the 2024 Joint International Conference on Computational Linguistics, Language Resources and Evaluation (LREC-COLING 2024), pages 2296–2306, Torino, Italia. ELRA and ICCL.
> >
> > [12] Wang, J., Xie, X., Hu, Q., Liu, S., Yu, J., Kong, J., & Li, Y. (2025, November). Defects4C: Benchmarking Large Language Model Repair Capability with C/C++ Bugs. In Proceedings of the 2025 IEEE/ACM International Conference on Automated Software Engineering (ASE).
> >
> > [13] Li, Rui, et al. "Optimizing code retrieval: High-quality and scalable dataset annotation through large language models." Proceedings of the 2024 Conference on Empirical Methods in Natural Language Processing. 2024.
> >
> > [14] Wang, Jianyou Andre, et al. "Scientific document retrieval using multi-level aspect-based queries." Advances in Neural Information Processing Systems 36 (2023): 38404-38419.
> >
> > [15] Wu, Siwei, et al. "Scimmir: Benchmarking scientific multi-modal information retrieval." Findings of the Association for Computational Linguistics: ACL 2024. 2024.
> >
> > ## Results Presentation
> >
> > We thank the reviewer for noting issues of clarity.. We will consolidate the experimental conditions from Tables 3 and 4 into a single table. To ensure readability, this new table will focus on the most critical metrics, following the similar format used for our fine-tuning experiments, which allows for direct comparison across settings. The full set of retrieval metrics will be moved to the appendix for reference.

---

### Official Review · Reviewer_mRpN · 2025-10-31

**Soundness:** 3
**Presentation:** 3
**Contribution:** 2
**Rating:** 4
**Confidence:** 4

**Summary:**

The paper introduces CLARC, a new benchmark for code search focused on C/C++. CLARC includes 1,245 compilable query-code pairs divided into groups of varying dependency complexity. It also provides several robustness settings, such as anonymized identifiers, randomized names, and compilation to Assembly or WebAssembly.

**Strengths:**

The motivation is solid and relevant. Focusing on C/C++ fills a clear gap in existing benchmarks. The dataset being fully compilable enhances reproducibility. The robustness settings are well-designed and informative. The automated data generation pipeline with statistical validation is technically sound. Experimental coverage is broad and clearly demonstrates the weakness of current models.

**Weaknesses:**

1. The dataset is small compared to existing large benchmarks (e.g., CodeSearchNet).
2. The contribution is mainly engineering rather than conceptual.
3. The analysis of why performance drops is shallow, with little insight into model behavior or representation.
4. The LLM-generated query validation focuses on surface quality rather than semantic fidelity.
5. The paper reads more like a dataset report than a research study, and the novelty is limited. Writing is clear but somewhat lengthy and repetitive.

**Questions:**

1. Have the authors tried fine-tuning models on CLARC to test whether robustness can be learned?
2. Can the anonymization and randomization pipelines be extended to other programming languages?
3. Is there any observed correlation between code complexity (e.g., cyclomatic complexity) and robustness degradation?

---

> ### Author Response · Authors · 2025-11-20
>
> We sincerely thank the reviewer for the thoughtful and constructive feedback. We respond to each point below.
>
> ## Dataset extension and model finetuning
>
> To demonstrate CLARC’s extensibility, we crawled 99 additional C/C++ GitHub repositories to construct a training set containing approximately 5,400 (query, code snippet) pairs. We then finetuned CodeT5+ and OASIS on this newly collected dataset using the InfoNCE loss with in-batch negatives[1] for 5 epochs. The models were evaluated under standard, neutralized, and randomized settings. Table 1 reports the performance of these finetuned models.
>
> **Table 1: Performance of the Finetuned Models. All reported metrics are Mean Reciprocal Rank (MRR) in percentage due to space constraints.**
>
> | | | | Group 1 |  | |  | Group 2 |  | |  | Group 3 |  |
> |---|---|:---:|:---:|:---:|---|:---:|:---:|:---:|---|:---:|:---:|:---:|
> | | FT Setting | Standard | Neutralized | Randomized | | Standard | Neutralized | Randomized | | Standard | Neutralized | Randomized |
> | **CodeT5+** | W/o ft. | 58.8 | 40.2| 29.5| | 46.7| 15.7| 11.3| | 17.8| 5.21| 5.18|
> | | ft. on Std. | 76.5 | 70.2 | 50.7 | | 67.0 | 31.9 | 23.8 | | 41.9 | 15.0 | 9.1 |
> | | ft. on Neu | 74.5 | 72.7 | 52.3 | | 60.6 | 38.7 | 24.1 | | 37.9 | 20.4 | 10.3 |
> | | ft. on Ran. | 75.9 | 70.0 | 60.9 | | 60.0 | 33.8 | 32.0 | | 39.5 | 17.2 | 11.7 |
> | **OASIS** | W/o ft. | 86.5| 83.7| 78.7| | 88.3| 68.9| 60.2| | 63.5| 36.1| 30.5|
> | | ft. on Std. | 85.0 | 79.7 | 72.3 | | 78.3 | 54.9 | 45.6 | | 57.5 | 24.4 | 16.0 |
> | | ft. on Neu | 87.5 | 83.4 | 77.6 | | 76.0 | 53.3 | 45.8 | | 57.9 | 37.3 | 21.2 |
> | | ft. on Ran. | 82.8 | 79.1 | 77.2 | | 68.2 | 46.4 | 45.4 | | 51.0 | 32.2 | 15.5 |
>
>
> The results reveal distinct behaviors between CodeT5+ and OASIS following finetuning. CodeT5+ demonstrates significant MRR improvements across all test settings (standard, neutralized, and randomized). This suggests that although the original CodeT5+ was exposed to C++ sources during pretraining [2], it was not fully adapted to the format, and our finetuning effectively bridges this gap. Conversely, OASIS appears already well-adapted to C/C++ code; thus, finetuning yields minimal gains. In fact, when finetuning on standard or neutralized data, OASIS maintains performance comparable to the pretrained checkpoint on the target setting but degrades on others. Notably, finetuning on the randomized setting leads to overall performance degradation for OASIS.
>
> Crucially, regardless of the training data used, a distinct performance gap remains between the standard setting and the neutralized/randomized settings, indicating that improving the robustness of code search models is a non-trivial task that cannot be solved simply by training on certain data formats. We hope these findings encourage the community to place greater emphasis on the robustness of current code search models.
>
> [1] Tianyu Gao, Xingcheng Yao, and Danqi Chen. 2021. SimCSE: Simple Contrastive Learning of Sentence Embeddings. In Proceedings of the 2021 Conference on Empirical Methods in Natural Language Processing
>
> [2] Wang, Yue, et al. "Codet5+: Open code large language models for code understanding and generation." Proceedings of the 2023 conference on empirical methods in natural language processing. 2023.
>
>
> ## LLM-generated query validation
>
> We clarify that semantic correctness is the primary scoring criterion: any mismatch immediately results in a -1 score. Therefore, the 95% CI in Table 2 reflects semantic rigor rather than surface fluency.
>
> Additionally, our human validation set covers over 30% of all data points in the current version of CLARC. To ensure robustness and reliability, we strategically focused our annotation efforts on the most challenging samples, with 50% of all Group 3 data points being manually validated. This level of validation meets or exceeds standards established in recent related work. For instance, [1] manually assessed 200 samples for a similar code retrieval task focusing on Python, while broader LLM-annotated tasks [2, 3] validated 250 and 100 samples, respectively, on datasets larger than CLARC.
> We will also release the human-written queries from this validation set, creating a subset containing only natural queries. This will address potential concerns about the homogeneity of LLM-generated queries.
> [1] Li, Rui, et al. "Optimizing code retrieval: High-quality and scalable dataset annotation through large language models." Proceedings of the 2024 Conference on Empirical Methods in Natural Language Processing. 2024.
>
> [2] Wang, Jianyou Andre, et al. "Scientific document retrieval using multi-level aspect-based queries." Advances in Neural Information Processing Systems 36 (2023): 38404-38419.
>
> [3] Wu, Siwei, et al. "Scimmir: Benchmarking scientific multi-modal information retrieval." Findings of the Association for Computational Linguistics: ACL 2024. 2024.

---

> > ### Author Response · Authors · 2025-11-20
> >
> > ## Additional correlation analysis on model robustness
> >
> > To investigate the specific features correlated with model robustness, we performed additional experiments. Directly analyzing retrieval metrics is challenging because they are confounded by the similarity between the query and candidate code snippets, making them an indirect measure of robustness. Instead, we focused our analysis on shifts in the code snippet embeddings, quantified by the $L_2$ distance between the embeddings.
> >
> > Formally, let $v_o$ be the embedding of the original code, $v_n$ be the embedding of the neutralized code (where identifiers are replaced by generic placeholders like `var_a`, `func_b`), and $v_r$ be the embedding of the randomized code (where identifiers are replaced by random character sequences). We analyze the correlation between Line of Code, Cyclomatic Complexity, Perturbation Fraction, and the **embedding shift distances** ($\|v_o - v_r\|_2$ and $\|v_o - v_n\|_2$).
> >
> > **Table 2: Correlation between the embedding shift distance and various features.**
> > | Model | Group1 Neutralized | Group1 Randomized | Group2 Neutralized | Group2 Randomized | Group3 Neutralized | Group3 Randomized |
> > | :--- | :---: | :---: | :---: | :---: | :---: | :---: |
> > | **Line of Code** | | | | | | |
> > | CodeT5+ | -0.113 | -0.131 | -0.115 | -0.196 | -0.025 | 0.080 |
> > | OASIS | -0.148 | 0.084 | -0.271 | -0.008 | 0.072 | -0.298 |
> > | NOMIC | -0.195 | -0.234 | -0.311 | -0.099 | -0.069 | -0.346 |
> > | Voyage | -0.308 | -0.321 | -0.341 | -0.390 | -0.303 | -0.275 |
> > | **Cyclomatic Complexity**| | | | | | |
> > | CodeT5+ | 0.043 | -0.044 | -0.124 | -0.103 | -0.158 | -0.070 |
> > | OASIS | -0.081 | -0.113 | -0.117 | 0.141 | 0.017 | 0.081 |
> > | NOMIC | -0.011 | -0.231 | -0.173 | 0.013 | -0.120 | -0.037 |
> > | Voyage | -0.212 | -0.191 | -0.165 | -0.198 | -0.226 | -0.191 |
> > | **Perturbation Fraction**| | | | | | |
> > | CodeT5+ | 0.335 | 0.515 | 0.402 | 0.523 | 0.371 | 0.193 |
> > | OASIS | 0.762 | 0.342 | 0.596 | 0.632 | 0.375 | 0.369 |
> > | NOMIC | 0.442 | 0.612 | 0.635 | 0.687 | 0.525 | 0.534 |
> > | Voyage | 0.751 | 0.740 | 0.663 | 0.702 | 0.621 | 0.653 |
> >
> > As shown in the results, both Line of Code (LOC) and Cyclomatic Complexity (CC) **show little correlation with** the embedding shifts. In most cases, we observe a weak **negative** correlation: as code length or complexity increases, the embedding shift decreases (i.e., embeddings for complex code are slightly more stable against identifier perturbation). While the magnitude of this correlation is higher for Voyage-code-3, it generally remains weak ($|r| < 0.4$).
> >
> > A significantly stronger predictor of embedding shifts is the **Perturbation Fraction**. We define this feature as the ratio of the cumulative length of all modified identifiers to the total length of the code snippet. Across all code groups and models, this fraction exhibits a positive correlation with embedding changes, and the magnitude of the correlation is much stronger compared to LOC and CC. **This result aligns with our hypothesis** that the models’ embeddings are highly sensitive to surface-level lexical overlap rather than code semantic understanding: as a larger proportion of the code's tokens are altered (even while preserving structural logic), the resulting embedding diverges more significantly from the original vector space.
> >
> >
> >
> > ## Generalizability of anonymization and randomization to other languages
> >
> > Yes, our anonymization and randomization pipeline can be easily extended to other programming languages. This would simply require using a widely available parser specific to the target language to locate identifiers such as variables and functions.

---

### Official Review · Reviewer_abmH · 2025-11-05

**Soundness:** 3
**Presentation:** 3
**Contribution:** 3
**Rating:** 6
**Confidence:** 4

**Summary:**

This paper introduces CLARC (C/C++ LAnguage Retrieval with Anonymized Code), a benchmark designed to evaluate the robustness of code search models on C/C++ code. The dataset consists of 1,245 query-code pairs and is categorized based on the complexity of the code dependencies. CLARC includes configurable settings like anonymized identifiers and low-level code representations (Assembly and WebAssembly) to test models across varying abstraction levels. The authors evaluate six state-of-the-art code search methods and demonstrate significant performance degradation when identifiers are anonymized or code is compiled into lower-level languages. Furthermore, the paper introduces an automated pipeline for scalable benchmark generation, making the dataset reusable and extensible for future work.

**Strengths:**

1. CLARC is a **novel benchmark** specifically for C/C++ code search, with unique settings for anonymized identifiers and low-level languages.
2. The experimental design is thorough, with a wide range of models tested across different evaluation settings. The inclusion of low-level language scenarios is a valuable addition.
3. The paper is mostly clear and well-structured, with good use of figures and tables to explain the methodology and results.
4. The work addresses a gap in code search research, particularly for industrial programming languages like C/C++, and the automated pipeline for benchmark generation has broad potential for future research.

**Weaknesses:**

1. While CLARC is comprehensive, the authors could explore **more complex real-world codebases** to further ensure the dataset's robustness.
2. While the automated benchmark generation pipeline is promising, the paper could offer a more in-depth discussion of how well this approach can scale to other programming languages or larger codebases.
3. The evaluation of low-level languages (Assembly, WebAssembly) is insightful but could benefit from a deeper analysis of why models struggle with such code (e.g., complexity of instruction sets, abstraction loss).

**Questions:**

1. Could the authors provide more details on the **scalability** of the automated benchmark generation pipeline, particularly when extended to other languages beyond C/C++?
2. The paper discusses performance degradation when identifiers are anonymized. Would the authors consider testing **semantic-based anonymization** (e.g., anonymizing function names based on their role) to better assess the models' understanding of code semantics?

---

> ### Author Response · Authors · 2025-11-20
>
> We sincerely thank the reviewer for the thoughtful and constructive feedback. We respond to each point below.
>
> ## Scalability of CLARC
>
> ### Extend to additional C/C++ repositories
>
> We appreciate the reviewer's question regarding scalability. Our pipeline already supports scaling to additional real-world repositories. To demonstrate this, we used our workflow to crawl 99 additional C/C++ GitHub repositories and constructed a new training set containing approximately 5,400 (query, code snippet) pairs. Table 1 summarizes the statistics.
>
> **Table 1: Training set statistics**
> | # of Samples | # of Tokens in Query | # of Tokens in Code | LOC  | CC  |
> |--------------|----------------------|---------------------|------|-----|
> | 5472         | 85.6                 | 263.8               | 31.2 | 4.6 |
>
> We subsequently fine-tuned CodeT5+ on this expanded dataset and evaluated it against our original benchmark. As shown in Table 2, the fine-tuned model achieves substantial performance improvements over the baseline across all groups. These positive results not only demonstrate the extendability of our workflow but also strongly validate the quality of the extended training set, as the model was able to effectively learn and generalize from the new data.
>
> We will highlight these scalability results in the paper.
>
> **Table 2: Performance of the finetuned CodeT5+**
> | | | | Group 1 |  | |  | Group 2 |  | |  | Group 3 |  |
> |---|---|:---:|:---:|:---:|---|:---:|:---:|:---:|---|:---:|:---:|:---:|
> | | FT Setting | Standard | Neutralized | Randomized | | Standard | Neutralized | Randomized | | Standard | Neutralized | Randomized |
> | **CodeT5+** | W/o ft. | 58.8 | 40.2| 29.5| | 46.7| 15.7| 11.3| | 17.8| 5.21| 5.18|
> | | ft. on Std. | 76.5 | 70.2 | 50.7 | | 67.0 | 31.9 | 23.8 | | 41.9 | 15.0 | 9.1 |
> | | ft. on Neu | 74.5 | 72.7 | 52.3 | | 60.6 | 38.7 | 24.1 | | 37.9 | 20.4 | 10.3 |
> | | ft. on Ran. | 75.9 | 70.0 | 60.9 | | 60.0 | 33.8 | 32.0 | | 39.5 | 17.2 | 11.7 |
>
>
> Nevertheless, we respectively note that the current version of CLAR already demonstrates a high standard of quality. As noted on our benchmark's Hugging Face page (https://huggingface.co/datasets/ClarcTeam/CLARC), the current dataset draws from high-quality, real-world projects: *86.7%* of source repositories have more than 100 GitHub stars, and *57.8%* have more than 500 stars, spanning domains such as data science, web, networking, and graphics. We will highlight the representativeness in the revised manuscript.
>
> When extending CLARC to code that differs substantially in complexity distribution, we could rerun the same hypothesis-testing and expert review to ensure new code is aligned with the existing benchmark's semantic rigor while preserving scalability.
>
> In summary, our pipeline provides robust support for extending CLARC with more real-world repositories. Validating the inclusion of code with fundamentally different complexity requires additional hypothesis testing.
>
> ### Extend to other programming languages
>
> We focused on C/C++ for two main reasons. First, as mentioned in our paper (L54-56), high-quality code search benchmarks for C/C++ are rare. Second, C/C++ code can be readily converted to other low-level languages, such as Assembly and Wasm, which enables further research directions.
>
> Meanwhile, our pipeline is fundamentally language-agnostic. Extending it to Java, C#, or other languages require only (1) sampling new, representative code snippets for the target language and (2) conducting the corresponding annotation and hypothesis testing. The primary constraint for full extensions during the rebuttal period is securing expert annotators. We will clarify this in the revised paper.

---

> > ### Author Response · Authors · 2025-11-20
> >
> > ## Semantic-based anonymization
> > We thank the reviewer for the insightful suggestion. Our current anonymization experiments were primarily designed to isolate the extent to which models rely on superficial lexical features (identifiers), as these are common targets for adversarial attacks [1, 2]. However, we believe the Assembly and Wasm settings within our CLARC framework already serve as a rigorous proxy for this: they represent an aggressive form of semantic-based anonymization where original identifiers are removed, but the code’s semantic logic is strictly preserved. The significant performance drop we observed in these settings reinforces our finding that models struggle when deprived of lexical cues. We agree that role-based renaming is a valuable future direction and will add a discussion to the Future Work section.
> >
> > ## Deeper analysis of why models struggle with low-level languages
> >
> > To understand why models degrade on low-level languages, we analyzed the structural characteristics of the code across all three CLARC groups. We measured the average Lines of Code (LOC), token count, and the number of unique keywords/instructions for the Standard, Assembly, and Wasm settings.
> >
> > **Table 3: Comparison between Standard, Assembly, and Wasm Settings**
> > | Metric | Group1 Standard | Group1 Asm | Group1 Wasm | Group2 Standard | Group2 Asm | Group2 Wasm | Group3 Standard | Group3 Asm | Group3 Wasm |
> > | :--- | :--- | :--- | :--- | :--- | :--- | :--- | :--- | :--- | :--- |
> > | LOC | 12.8 | 80.7 | 96.2 | 13.3 | 84.4 | 134.4 | 71.5 | 212.3 | 138.3 |
> > | # of Tokens | 119.2 | 753.7 | 665.5 | 137.7 | 831.3 | 947.1 | 706.9 | 2272.6 | 967.8 |
> > | # Unique keywords/Instructions | 4.77 | 16.46 | 13.61 | 4.53 | 15.39 | 13.02 | 6.62 | 21.48 | 15.44 |
> >
> >
> > As shown in Table 3, compiling to Assembly or Wasm results in a 5×-10× increase in LOC and a significant expansion in token usage (e.g., Group 3: ~706 -> ~2272 tokens). Additionally, the rise in unique keywords/instructions indicates a more complex instruction set compared to standard syntax. These factors likely overwhelm the models' ability to capture semantic equivalence, as the dense logic of high-level code is diluted across hundreds of low-level instructions. Unfortunately, the black-box nature of current state-of-the-art models prevents a deeper mechanistic analysis of these failures. We cannot inspect internal attention weights or embeddings to pinpoint exactly where semantic alignment breaks down during the processing of such verbose sequences. This limitation underscores the contribution of our work: CLARC offers a pipeline for automatically generating aligned datasets for low-level languages. We envision CLARC facilitating the training and evaluation of white-box models on these languages, enabling more transparent, layer-wise analysis.
> >
> > [1] Yefet, Noam, Uri Alon, and Eran Yahav. "Adversarial examples for models of code." Proceedings of the ACM on Programming Languages 4. OOPSLA (2020): 1-30.
> >
> > [2] Hort, Max, Linas Vidziunas, and Leon Moonen. "Semantic-Preserving Transformations as Mutation Operators: A Study on Their Effectiveness in Defect Detection." 2025 IEEE International Conference on Software Testing, Verification and Validation Workshops (ICSTW). IEEE, 2025.

---

> ### Comment · Reviewer_abmH · 2025-11-27
>
> Thank you for the thorough rebuttal. My concerns have been resolved, and I will keep my current rating.

---

### Meta-Review · Area_Chair_3PZs · 2025-12-09

**Summary:**

The paper introduces a benchmark focused on code search from text queries. The data is taken from github, and textual queries are synthetically generated. More importantly, the dataset is grouped across different dimensions enabling evaluation across varying code complexity and underlying languages. Performance under anonymisation of identifiers can also be assessed, to control for the reliance of encoders in simple lexical matching rather than semantic similarity. The dataset is relevant and useful to the community, and so is the evaluations reported by the authors covering state-of-the-art encoders and highlighting their limitations.

**Reviewer Concerns:**

Reviewers mostly raised concerns related to the relevance of code search and of the underlying programming languages represented in the data. Besides that, comments were made regarding the quality assessment of queries and scalability of the data producing pipeline. Authors rebutted adequately in my opinion showing evidence they could scale the data generation, and presenting details on the coverage of the data quality assessment.

**Reviewer Scores:**

Reviewer abmH would have maintained their origin score of 6.
Reviewer eTqq wasn't convinced by the rebuttal and would likely maintain a score of 4, to which I disagree as their concerns were mostly related to relevance of the data and problem. I believe authors rebutted adequately and the problem and chose languages are indeed quite relevant.
Reviewer mRpN had an original score of 4 and, in my view, had their concerns well addressed by the authors, so I'd expect an increase in score.

---

### Decision · Program_Chairs · 2026-01-26

Accept (Poster)